# Gastrointestinal helminths increase *Bordetella bronchiseptica* shedding and host variation in supershedding

**Nhat TD Nguyen[1,2†], Ashutosh K Pathak[1,2,3†], Isabella M Cattadori[1,2]\***

[1]Center for Infectious Disease Dynamics, The Pennsylvania State University, University Park, United States; [2]Department of Biology, The Pennsylvania State University, University Park, United States; [3]Department of Infectious Diseases, University of Georgia, Athens, United States

**Abstract** Co-infected hosts, individuals that carry more than one infectious agent at any one time, have been suggested to facilitate pathogen transmission, including the emergence of super-shedding events. However, how the host immune response mediates the interactions between co-infecting pathogens and how these affect the dynamics of shedding remains largely unclear. We used laboratory experiments and a modeling approach to examine temporal changes in the shedding of the respiratory bacterium *Bordetella bronchiseptica* in rabbits with one or two gastrointestinal helminth species. Experimental data showed that rabbits co-infected with one or both helminths shed significantly more *B. bronchiseptica*, by direct contact with an agar petri dish, than rabbits with bacteria alone. Co-infected hosts generated supershedding events of higher intensity and more frequently than hosts with no helminths. To explain this variation in shedding an infection-immune model was developed and fitted to rabbits of each group. Simulations suggested that differences in the magnitude and duration of shedding could be explained by the effect of the two helminths on the relative contribution of neutrophils and specific IgA and IgG to *B. bronchiseptica* neutralization in the respiratory tract. However, the interactions between infection and immune response at the scale of analysis that we used could not capture the rapid variation in the intensity of shedding of every rabbit. We suggest that fast and local changes at the level of respiratory tissue probably played a more important role. This study indicates that co-infected hosts are important source of variation in shedding, and provides a quantitative explanation into the role of helminths to the dynamics of respiratory bacterial infections.

**\*For correspondence:**
imc3@psu.edu

†These authors contributed equally to this work

**Competing interest:** The authors declare that no competing interests exist.

## Editor's evaluation

The authors perform experimental infections with rabbits to study how coinfection with one or more helminths affects the shedding of the respiratory bacterium *Bordetella bronchiseptica*. The results show that shedding varies strongly from one individual to the next and that co-infections with helminths lead to increased levels of shedding. The authors nicely combine within-host kinetics modelling and their longitudinal data to estimate key parameter values associated with bacterium and immune growth rates in the four conditions. These suggest that the shedding differences can be explained by differences in bacterial growth.

## Introduction

Individual variation in pathogen transmission can increase the basic reproduction number $R_0$ of a pathogen and determine whether an infection will invade and spread or stutter and quickly fade out

in a population of susceptible hosts (*Keeling et al., 2001*; *Lloyd-Smith et al., 2005*). One of the causes of this variation is associated with differences in the amount and duration of pathogen shedding, whereby some infected hosts shed disproportionately more and for longer than the average population, the so called supershedders (*Chase-Topping et al., 2008*; *Gopinath et al., 2013*), while others do not shed at all (*Chen et al., 2006*; *Hadinoto et al., 2009*; *Leung et al., 2015*). Co-infected hosts are frequently proposed to contribute to this variation, which could emerge from interactions between pathogen species mediated by the host immune response and the consequences on host infectiousness (*Sheth et al., 2006*; *Graham et al., 2007*; *Richard et al., 2014*).

While studies on the immunological response to multi-species infections has provided insight to the interactions between the host and its pathogens, there remains a need to identify how these processes relate to onward transmission, specifically the patterns of bacterial shedding. Dynamical mathematical models are particularly useful in disentangling these complexities as they can generate mechanistic-driven hypotheses that can be examined in relation to empirical data (*Smith et al., 2013*; *Byrne et al., 2019*). In the current study, we applied this general approach to investigate the dynamics of shedding of the respiratory bacterium *Bordetella bronchiseptica* in rabbits experimentally co-infected with one or both of the gastrointestinal helminths *Trichostrongylus retortaeformis* and *Graphidium strigosum*. We explored to what extent helminths could alter the level of *B. bronchiseptica* shedding over time, whether the trend varied depending on the helminth species and to what extent the host immune response could explain the patterns observed.

*B. bronchiseptica* is a highly contagious bacterium of the respiratory tract that causes multiple symptoms and infects a wide range of mammals (*Goodnow, 1980*). In rabbits and mice, and likely other mammal species, *B. bronchiseptica* is removed from the lower respiratory tract (lungs and trachea) via phagocytosis stimulated by a Th1 inflammatory reaction that involves cell mediated antibodies and neutrophils (*Thakar et al., 2007*; *Thakar et al., 2009*; *Thakar et al., 2012*). Bacteria persist in the nasal cavity, although they are partially reduced by IgA antibodies in naïve and immunized mice (*Kirimanjeswara et al., 2003*; *Wolfe et al., 2007*). Transmission is poor among wild-type laboratory mice but increases among TLR4-deficient mice (*Rolin et al., 2014*). The TLR4-deficient response is associated with neutrophil infiltration, and the intensity of shedding has been found to be positively correlated with neutrophil counts. In contrast, rabbits naturally shed (*Pathak et al., 2010*) and efficiently transmit *B. bronchiseptica* between animals (*Long et al., 2010*). The evidence of rapid outbreaks in pet kennels, livestock holdings and laboratory rabbitries are consistent with rapid transmission among animals in close contact. Occasionally, *B. bronchiseptica* spills-over into humans but there is no evidence of sustained onward transmission, as these human infections have invariably occurred in immunocompromised individuals (*Goodnow, 1980*; *Woolfrey and Moody, 1991*). Humans are primarily infected by the species *B. pertussis* and *B. parapertussis* which are responsible for regular whooping cough outbreaks worldwide (*Domenech de Cellès et al., 2018*). Given the close relatedness between *Bordetella* species, and considering the many similarities in the kinetics of infection and the immune response, *B. bronchiseptica* provides a good model system to explore the dynamics of shedding in *Bordetella* infections and the interaction with other infectious agents, like gastrointestinal helminths.

In co-infections with other respiratory pathogens *B. bronchiseptica* contributes to exacerbate respiratory symptoms, including the development of acute pulmonary disease and bronchopneumonia, and ultimately host death (*Brockmeier et al., 2008*; *Loving et al., 2010*; *Schulz et al., 2014*; *Kureljušić et al., 2016*; *Hughes et al., 2018*). Parasitic helminths commonly stimulate a Th2 anti-inflammatory immune response that interferes with the Th1 response developed against *B. bronchiseptica* and related species (*Brady et al., 1999*; *Pathak et al., 2012*; *Thakar et al., 2012*). We recently showed that rabbits co-infected with either *T. retortaeformis* or *G. strigosum* carried higher *B. bronchiseptica* infections in the nasal cavity but there were no significant differences in the size and duration of infection in the lungs, when compared to rabbits infected with bacteria alone (*Pathak et al., 2010*; *Pathak et al., 2012*; *Thakar et al., 2012*). These two helminths stimulate a similar immune response, however, while the former is reduced or cleared from the small intestine, the latter persists with high intensities in the stomach (*Cattadori et al., 2005*; *Cattadori et al., 2008*; *Cattadori et al., 2019*). The modeling of the immune network in *B. bronchiseptica-T. retortaeformis* rabbits and *B. bronchiseptica* alone animals suggested that neutrophils and antibodies IgA and IgG are important to bacterial clearance from the lungs (*Thakar et al., 2012*). Most likely, this mechanism could also explain patterns observed

in *B. bronchiseptica*-*G. strigosum* infections where bacteria are removed from the lower but not the upper respiratory tract (*Pathak et al., 2012*). Laboratory infections of mice with *B. bronchiseptica* supported the role of these three immune variables to bacterial clearance from lungs and trachea, and reduction in the nasal cavity (*Kirimanjeswara et al., 2003*; *Wolfe et al., 2007*).

To investigate the impact of these two helminths to *B. bronchiseptica* shedding, four types of laboratory infections were investigated: i- *B. bronchiseptica* (B) only, ii- *B. bronchiseptica*-*G. strigosum* (BG), iii- *B. bronchiseptica*-*T. retortaeformis* (BT), and iv- *B. bronchiseptica*-*T. retortaeformis*-*G. strigosum* (BTG). For the first three experiments, studies on the dynamics of infection and related immune responses are available in *Pathak et al., 2010*; *Pathak et al., 2012*; *Thakar et al., 2012* and key findings have been described in the previous section. The triple infection (BTG) followed the same experimental design and laboratory procedures, and is presented here for the immune components relevant to this study, together with novel data on *B. bronchiseptica* shedding from all the four types of infection (details in Materials and methods). Longitudinal data on immunity and bacteria shedding over four months were then used to develop an individual-based Bayesian model that was applied to each type of infection, independently. First, we built a dynamical model that investigated the relative contribution of neutrophils and specific IgA and IgG to bacterial infection in the whole respiratory tract. Second, we applied this dynamical model to laboratory data, where shedding was linked to the estimated bacterial infection. Helminths were assumed to alter the immune response to *B. bronchiseptica* by impacting the magnitude and time course of the three immune variables. Simulations provided a parsimonious explanation of the dynamics of shedding and how the two helminth species, with contrasting dynamics, contributed to variation between types of infection and between individuals within the same infection.

## Results
### Experimental results: Helminth infections boost *B. bronchiseptica* shedding

For each type of infection, the number of bacteria shed, as determined by contact with a BG-blood agar petri dish for a fixed time (CFU/s), were collected every week or two-three times a week for every rabbit (see Materials and methods). There was large variation in the level at which *B. bronchiseptica* was shed both within and between the four types of infection (*Figure 1*). Some rabbits consistently shed large amounts of CFUs while others were low shedders or did not shed at all, specifically: 3 out of 16 for *B. bronchiseptica* only animals (B), 1 out of 23 for *B. bronchiseptica*-*G. strigosum* (BG), 3 out of 20 for *B. bronchiseptica*-*T. retortaeformis* (BT) and 0 out of 24 for the triple infection (BTG). Since we were interested in the dynamics of shedding, these non-shedders were excluded from the subsequent analyses. High shedders were found during the first 60 days post infection, although some of these rabbits continued to shed a large number of bacteria well beyond this time (*Figures 1 and 2*). The median trend of each group reflects the large number of events at medium-low intensity and the large variation among individuals observed. The level of shedding was significantly higher in rabbits co-infected with helminths than *B. bronchiseptica* only animals (mean shedding (CFU/s) and 95% CI, BTG: 0.42, 0.31–0.54; BG: 0.32, 0.25–0.41; BT: 0.26, 0.18–0.35 and B: 0.02,–0.01-0.03; Kruskal-Wallis test with Bonferroni correction for multiple testing, for all: p<0.05). When we considered the co-infected hosts, no significant differences were observed among the three groups (p>0.05) although the triple infection showed a tendency for higher shedding (*Figure 2*). Across the four infections up to 55% of the shedding events were classified as null, where rabbits interacted with the petri dish but did not shed even though they were known to be infected (*Figure 3*).The highest percentage was found in the *B. bronchiseptica* only group (56.2%), followed by the dual infections (BT: 40.3% and BG: 29.8%) and last the triple infection BTG (25.8%) (*Figure 3*). This latter group showed the largest variation in the level of shedding (Negative Binomial *k* and 95% CI, BTG: 0.21, 0.16–0.26) followed by the dual infections BG (0.37, 0.29–0.45) and BT (0.45, 0.33–0.57); for all, the distribution was not significantly different from the frequency data (p>0.05). There were only three bins to represent the frequency of shedding in the *B. bronchiseptica* alone group and the Negative Binomial distribution could not be fitted. These findings suggest that the presence of helminths increased the level of bacteria shed in the environment and the frequency of these events. Having helminths also increased the variation in the magnitude of these events among rabbits within the same group as well as between groups.

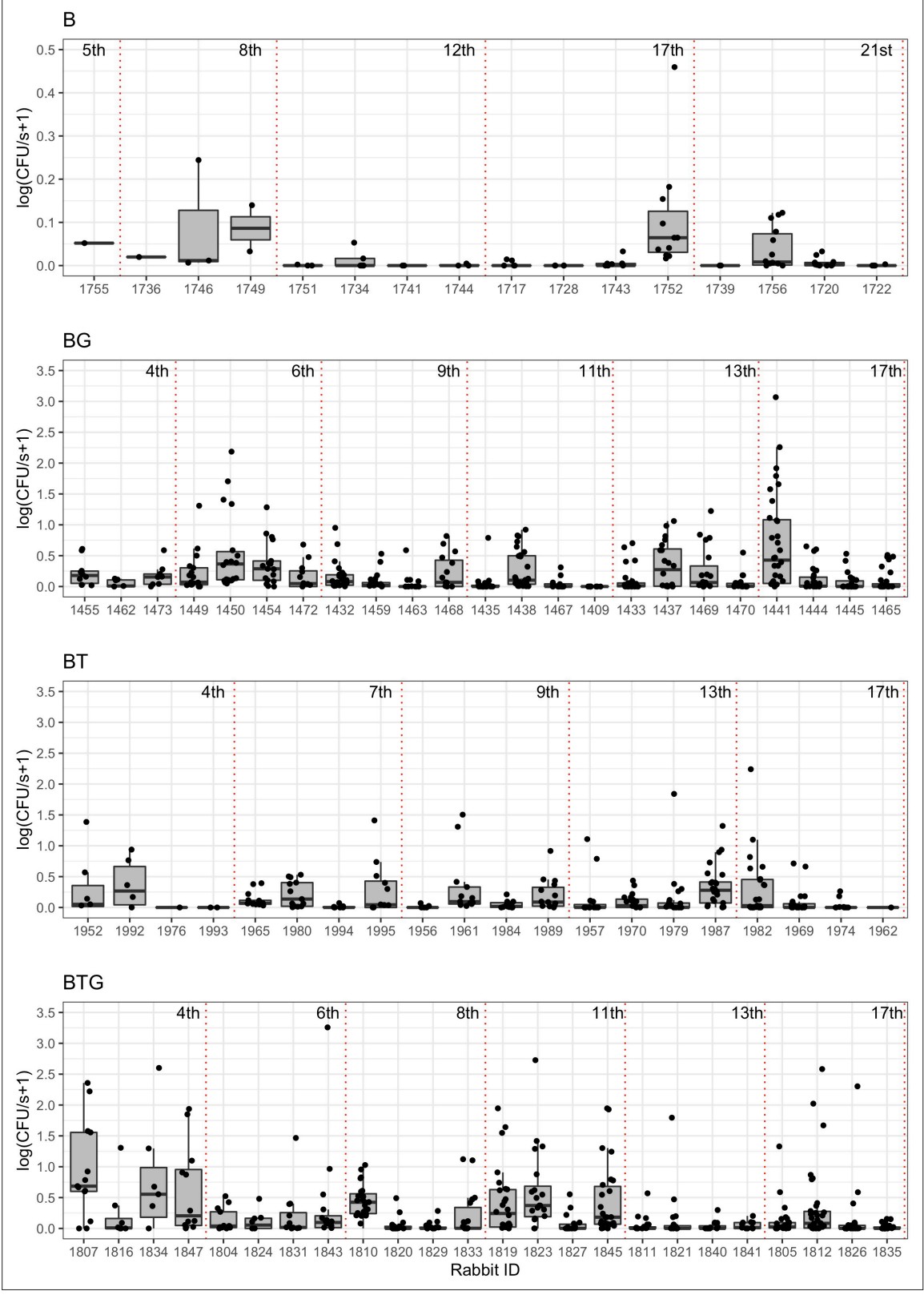

**Figure 1.** Observed levels of *B. bronchiseptica* shedding by contact of infected rabbits with a Bordet-Gengou agar petri dish. Rabbits (infected shedders and non-shedders) from the four types of infection are reported, *B. bronchiseptica* (B), *B. bronchiseptica-G. strigosum* (BG), *B. bronchiseptica-T. retortaeformis* (BT) and *B. bronchiseptica-T. retortaeformis-G. strigosum* (BTG) (top-down). Each infection is presented following the experimental design where rabbits are ordered by their time of sacrifice (early/left, late/right) during the infection (week 4th to 21st, dotted vertical red

*Figure 1 continued on next page*

*Figure 1 continued*

lines). For every rabbit the following are reported: observed shedding events (black points), 25th and 75th percentiles (top and bottom hinges), median values (thick horizontal lines) and values within the 1.5 inter-quartile range (IQR, whiskers extending up to 1.5*IQR). To facilitate the visualization, B infection is reported on a smaller y-axis. The level of shedding is quantified as number of Colony Forming Units per second (CFU/s).

Together these findings indicate that co-infections with helminths, irrespective of the species, should lead to a higher probability of *B. bronchiseptica* onward transmission given a contact.

## Experimental results: Helminth infections promote *B. bronchiseptica* supershedding

Few events were characterized by high bursts, very large numbers of CFUs counted in a petri dish (*Figures 2 and 4*). The definition of supershedder, namely, the threshold above which there is evidence of a supershedding event, depends on the pathogen and the host, and it is usually based on the assumption that hosts carry one single infection (*Chase-Topping et al., 2008*; *Gopinath et al., 2012*). If we estimate this threshold using *B. bronchiseptica* only infected animals, and define supershedders as the hosts that have at least one shedding event above this threshold, for example the 99th percentile, our cut-off value is CFU/s=0.27 (*Figure 3*). This indicates that none of the rabbits with *B. bronchiseptica* (0 out of 13) can be classified as supershedders. However, if we apply the same cut-off to the co-infected groups the percentage is considerably high, namely: 59% (13 out of 22) for BG, 53% (9 out of 17) for BT and 42% (10 out of 24) for the BTG group. This threshold definition provides a common reference value for our infections, however, it is dependent on host status (i.e. calculated using hosts with single infections), and is not representative in other settings, such that it overestimates the number of supershedders in rabbits that are co-infected. A way to overcome some of the limitations of this approach is to estimate the cut-off value using the whole dataset, irrespective of their type of infection. When we apply this approach the 99th percentile threshold is now CFU/s=8.58 and the fraction of hosts with, at least, one supershedding event above this value is: 0% (0 out of 13) for B, 32% (7 out of 22) for BG, 18% (3 out of 17) BT and 21% (5 out of 24) for BTG (*Figure 3*). Similarly, if we select the less limiting 95th percentile threshold, the new common value decreases to CFU/s=2.71 and the number of rabbits with at least one supershedding event is still zero for B but increases to 45% (10 out of 22) for BG, 29% (5 out of 17) for BT and 29% (7 out of 24) for BTG (*Figure 3*). Consistent with the general pattern of shedding, supershedding events were found more often in the initial four/five weeks post infection (*Figures 1 and 2*), although later events were also observed, especially for the BG and BTG groups, and with some rabbits contributing multiple times. Importantly, since co-infected rabbits have higher levels of shedding and a much higher probability of becoming supershedders (whether this is at the 95th or 99th percentile threshold), we should expect a higher probability of onward transmission given a contact during these disproportionate events, compared to *B. bronchiseptica* only rabbits or rabbits that have average values of shedding.

## Modeling results: Changes in the relative role of neutrophils and antibodies contribute to variation in *B. bronchiseptica* dynamics of infection between groups

We examined the hypothesis that *B. bronchiseptica* shedding is controlled by neutrophils, species-specific IgA and IgG produced against the bacterial infection in the respiratory tract, and helminths affect shedding by altering the magnitude and time course of the three immune variables. Our model formulation did not explicitly include the intensity of infection of the two helminths, rather, we examined how they impacted the immune response and its consequences on shedding. Here, we report on the relationships between immune response and dynamics of infection from the dynamical model fitted to the experimental individual data, while in the next section we examine the relationship between predicted level of infection and estimated intensity of bacteria shedding. To provide meaningful results, and to avoid issues associated with model convergence, individual time series with three or less shedding events larger than zero were omitted from the modeling. Similarly, for *B. bronchiseptica* only rabbits model fitting was performed on the data pooled at the group level because of the small number of hosts with enough shedding events larger than zero (details in Materials and methods). The description of the model framework, fitting and validation, including parameter

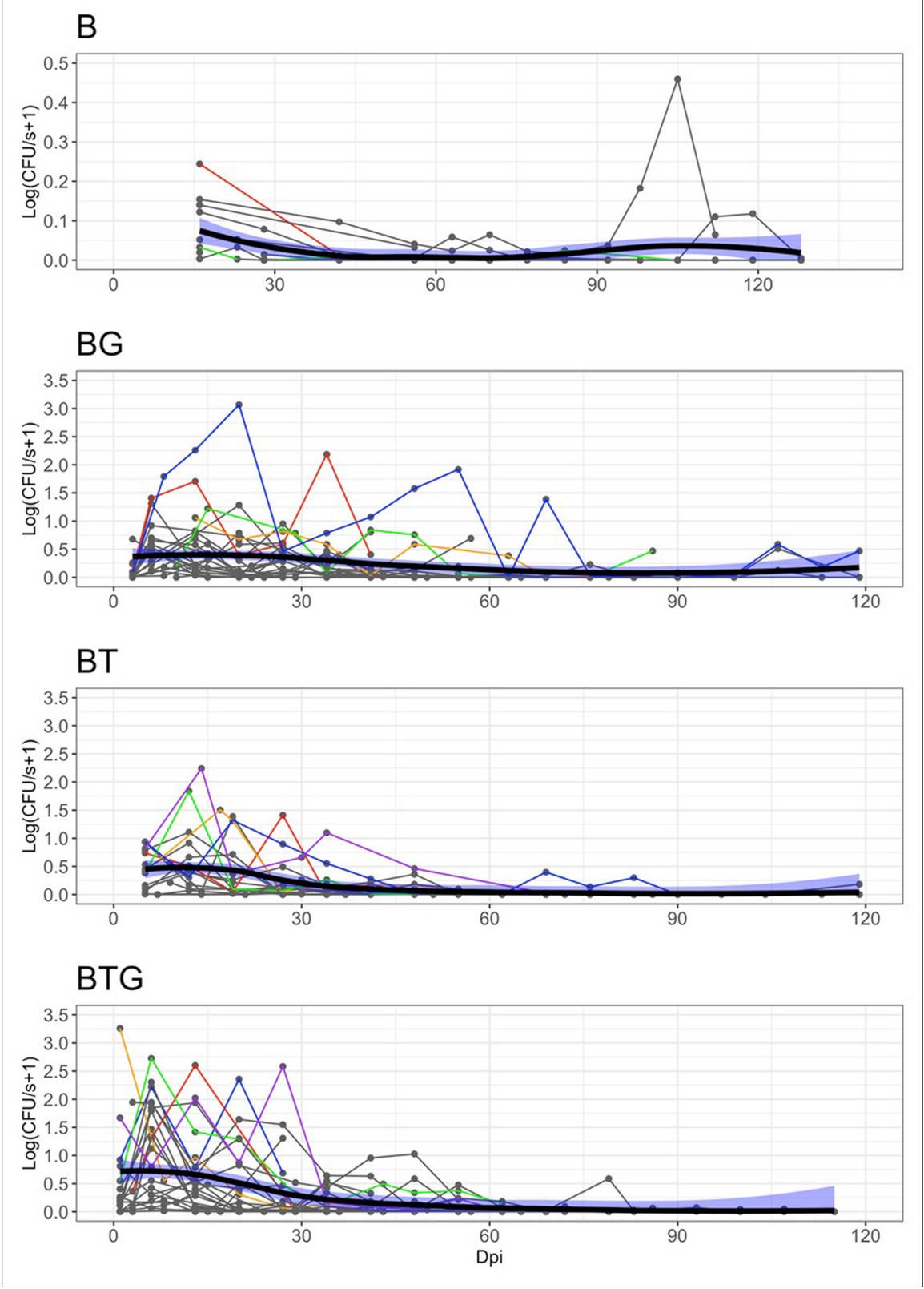

**Figure 2.** Observed level of *B.bronchiseptica* shedding by days post infection (Dpi) and type of infection. Shedding events (points), host individual trajectories (thin lines) and median population trend (smoothed thick lines) with 95% CI (blue shadows) are reported. The trajectories of few individuals are highlighted with different colors to facilitate visualization. The abrupt interruption of individual trends is caused by the removal of rabbits at fixed time points. To reduce overcrowding, host trajectories with at least four shedding points larger than zero are included. Note the much lower shedding values, and different y-axis, for the B group. Further details in *Figure 1*.

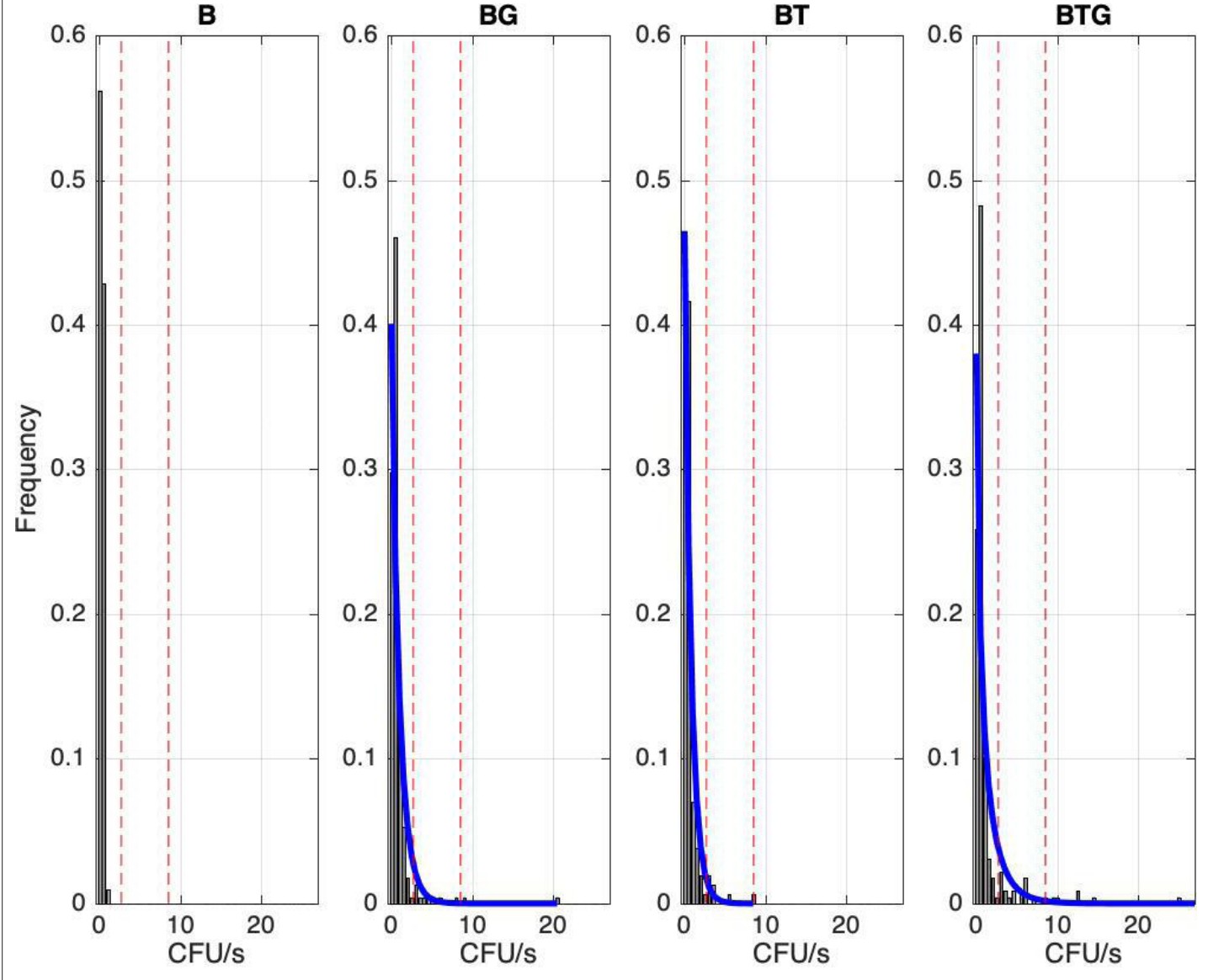

**Figure 3.** Frequency of observed *B.bronchiseptica* shedding events for the four types of infection. Events have been grouped by shedding level using 0.5 CFU/s unit intervals, the first interval represents events with zero shedding. Red lines indicate the 95th and 99th percentile thresholds, respectively, estimated using the entire experimental dataset; cases above the threshold value represent supershedding events. A Negative Binomial distribution is fitted to each co-infection group (blue line), while it was not possible for the B group due to the few bins.

calibration, model selection and sensitivity analysis of key parameters, are outlined in Materials and methods and Appendix. Consistently among the four types of infection, simulations indicated that there was a rapid bacterial replication following the initial inoculum, the growth rate, *r*, was high for BTG (group level value and [95% CI]: 1.47 [1.10–1.83]) and BG (1.18 [0.83–1.54]), less so for BT (1.01 [0.83–1.18]) and much lower for B single infection (0.06 [0.006–0.12]) (*Tables 1 and 2*). The growth of *B. bronchiseptica* prompted a rapid immune response. Model fitting well captured the empirical trends of neutrophils and specific antibodies over time, both as a profile representative of each type of infection and as a trend of every rabbit within each group (*Table 2*, *Figure 5*). In all four groups, neutrophils showed the fastest rate of increase, $a_1$, with a tendency to be faster in *B. bronchiseptica* only rabbits, followed by specific IgA and then specific IgG (*Table 2*, *Figure 6*). Neutrophils were also the fastest to decrease, at a rate $b_1$, and to return to original values with the neutralization, albeit no complete clearance, of bacteria from the respiratory tract (*Figures 5 and 6*). As reported in our previous studies of this system, *B. bronchiseptica* is removed from lungs and trachea but persists

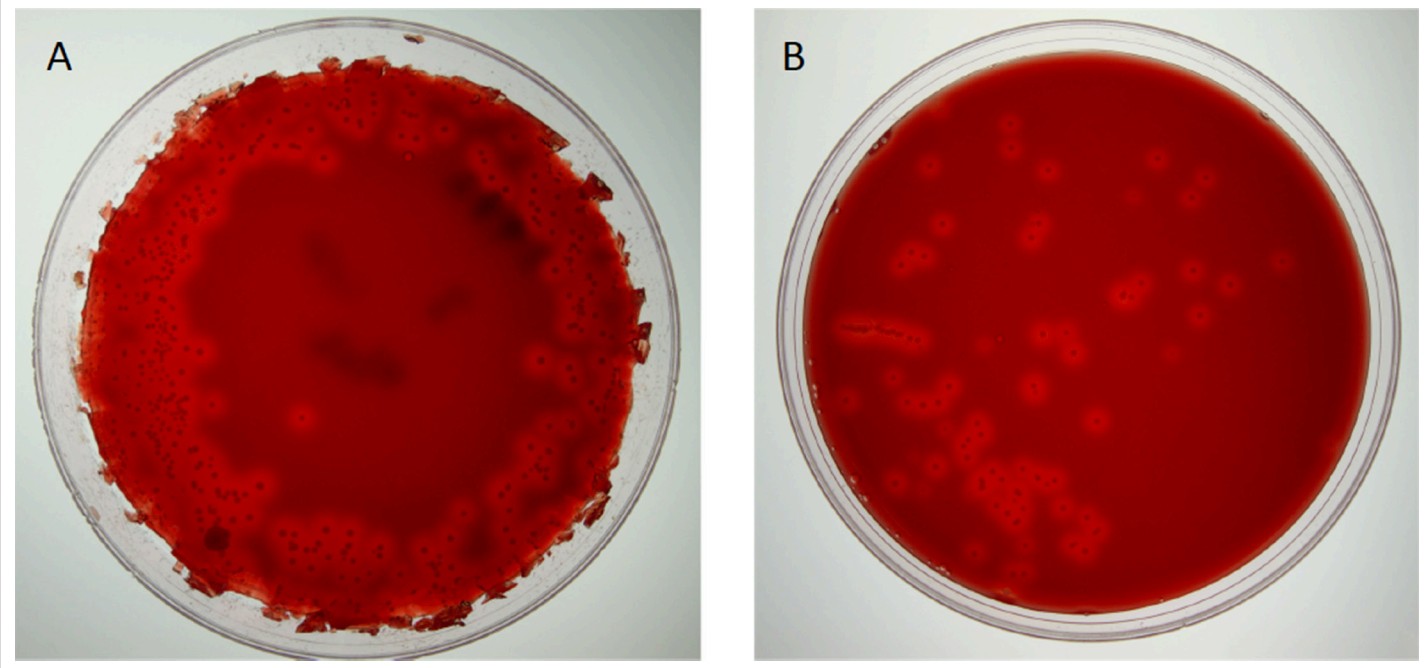

**Figure 4.** *B. bronchiseptica* shedding on BG-blood agar petri dishes. Examples of A: supershedding event and B: average shedding event.

in the nasal cavity (*Pathak et al., 2010*; *Pathak et al., 2012*; *Thakar et al., 2012*), a pattern also observed in mice (*Kirimanjeswara et al., 2003*). The increase of specific IgA and IgG was slow, however, once IgG reached the asymptotic value at around 20 days post infection, it remained high throughout the trials, while IgA peaked around day 20 and slowly decreased thereafter, although it had not fallen to baseline levels by the end of the trials (*Figures 5 and 6*). The relative contribution of the three immune variables to *B. bronchiseptica* neutralization, $c_1$, $c_2$, $c_3$, varied among the four types of infection. Model fittings suggested that neutrophils primarily controlled bacteria in the single infection, whereas a combination of neutrophils and specific IgG were probably important in BTG rabbits (*Table 2*, *Figure 6*). The coupling of IgG and IgA could contribute to explain the neutralization of bacteria in the BT group while neutrophils and IgG appeared to be relevant in the BG group. We note that timing of these interactions was also critical. Given the early reaction of neutrophils, they are expected to primarily control *B. bronchiseptica* early on in the infection, while specific IgG and, secondly, specific IgA appear to be more important at a later time, as they decreased more slowly, such is the case for IgA, or remained consistently high over the course of the infection, like for IgG (*Figures 5 and 6*). An investigation of the posterior parameter estimations confirmed that the immune response changed both between individuals within a group and between groups (Appendix 1).

## Modeling results: immune mediated *B. bronchiseptica* infection explains variation in the dynamics of shedding between groups

For each individual rabbit, we examined the dynamics of shedding by linking the empirical shedding of every individual to its estimated intensity of infection in the respiratory tract through the zero inflated log-normal function and, from this, generated the simulated individual shedding data (details in Materials and methods). As previously noted, model fitting was performed on the whole dataset for the *B. bronchiseptica* only rabbits. Simulations showed that the rapid growth of *B. bronchiseptica* in the respiratory tract led to the rapid shedding in the environment (*Figure 7*). The estimated time to reach the peak of shedding was between 10 and 12 days post infection for the co-infected groups, specifically: 9 days for BTG, 11 days for BT and 12 days for BG. For *B. bronchiseptica* only rabbits, technical problems with bacteria growing prevented us to collect shedding data in the first 10 days post infection, consequently, we missed to identify the peak of shedding. Model simulations place the peak right at the start of the trial, however, this should be taken with caution, since there are no

**Table 1.** Description and unit of parameters and variables used in the dynamical and observed models.

| Parameters/ | Descriptions | Unit | Value |
|---|---|---|---|
| Variables | | | |
| Rabbit $i^{th}$ specific parameters | | | |
| $a_{1,i}, b_{1,i}, c_{1,i}$ | Growth, decay and *B. bronchiseptica* neutralization rates of neutrophils | day$^{-1}$ | Estimated |
| $a_{2,i}, b_2, c_{2,i}$ | Growth, decay and *B. bronchiseptica* neutralization rates of IgA | day$^{-1}$ | Estimated |
| $a_{3,i}, b_3, c_{3,i}$ | Growth, decay and *B. bronchiseptica* neutralization rates of IgG | day$^{-1}$ | Estimated |
| $r_i$ | Per capita *B. bronchiseptica* replication rate | day$^{-1}$ | Estimated |
| $x_{1,i}(t)$ | Time-dependent neutrophil response | cells/ml | Calculated |
| $x_{2,i}(t)$ | Time-dependent IgA response | O.D. index | Calculated |
| $x_{3,i}(t)$ | Time-dependent IgG response | O.D. index | Calculated |
| $y_i(t)$ | Time-dependent *B. bronchiseptica* infection intensity | CFU/g count | Calculated |
| Group parameters | | | |
| $\mu a_1, \mu b_1, \mu c_1$ | Mean of neutrophil growth, decay and neutralization rates | day$^{-1}$ | Estimated |
| $\sigma a_1, \sigma b_1, \sigma c_1$ | Standard deviation (S.D.) of neutrophil growth, decay and neutralization rates | day$^{-1}$ | Estimated |
| $\mu a_2, \mu c_2$ | Mean of IgA growth, and neutralization rates | day$^{-1}$ | Estimated |
| $\sigma a_2, \sigma c_2$ | Standard deviation (S.D.) of IgA growth, and neutralization rates | day$^{-1}$ | Estimated |
| $\mu a_3, \mu c_3$ | Mean of IgG growth, and neutralization rates | day$^{-1}$ | Estimated |
| $\sigma a_3, \sigma c_3$ | Standard deviation (S.D.) of IgG growth, and neutralization rates | day$^{-1}$ | Estimated |
| $\mu r, \sigma r$ | Mean and S.D. of *B. bronchiseptica* replication rates | day$^{-1}$ | Estimated |
| $x_1^*$ | Neutrophils level at equilibrium | cells/ml | Observed |
| $\mu y_0$ | *B. bronchiseptica* infection intensity at time *t*=0 | CFU/g count | Estimated |

shedding data to train the model during the first 10 days (*Figure 7*). Therefore, while we still report the average trend of shedding during this period this has to be validated in the laboratory.

Results from model fitting confirmed that bacteria shedding reached higher levels and lasted for a longer time in co-infected rabbits than in hosts where *B. bronchiseptica* was the only infection (*Figure 7*). For example, the peak value (CFU/day; 95% CIs) was: 14,185 (11.06–12,258,022) for BTG, 8,266 (5.89–10,119,754) for BG and 7,863 (29.57–2,213,310) for BT. For completeness we also report the peak value for *B. bronchiseptica* only rabbits, which was 678 CFU/day (168–2723); however, this is not indicative of the real value, as already noted above. Model simulations well captured the different trends of shedding among the co-infections. Rabbits from the BTG infection exhibited the most rapid increase of shedding, the highest levels and the slowest decline post-peak, suggesting a possible synergistic effect of the two helminths. The BT group showed the most rapid decline, while rabbits from the BG group maintained a trend intermediate between the BTG and BT groups (*Figure 7*).

Model fitting of individual time series described well the general trend and the high levels of shedding early in the infection (*Figure 7* also compare it with *Figure 2*), particularly for BTG and BT. In contrast, the model fitted at the group level, showed a tendency to underestimate the average temporal profile, although we note that this is probably a visual effect. In fact, this trend reflected the large number of medium-low shedding events, including the fact that about 30% of these were classified as zero shedding and occurred throughout every co-infected group. The large variation in bacteria shed both within and between rabbits also contributed to this average pattern. Rabbits showed rapid changes in the level of shedding between consecutive samplings, where bursts of bacteria alternated to low or no shedding in a matter of a few days. This pattern was common among hosts, irrespective of their infection group. This rapid dynamic could not be explained by the intensity of infection and related immune responses at the scale used to record such data. Instead, it is possible that changes

**Table 2.** Model prior and posterior parameters for the immune response and *B. bronchiseptica* infection. We report: mean and S.D. of priors, and mean, 95% CI and S.D. from posterior distributions for each group. The complete description of the parameters is provided in *Table 1*.

| Parameters | Priors | Posteriors B | Posteriors BG | Posteriors BT | Posteriors BTG |
|---|---|---|---|---|---|
| $\mu_{a_1} * 10^{-4}$ | 0.10 (0.05) | 2.89 (1.17–4.61) | 1.08 (0.31–1.85) | 3.32 (1.73–4.90) | 0.93 (0.35–1.50) |
| $\mu_{a_2} * 10^{-5}$ | 1.10 (0.20) | 2.00 (1.65–2.35) | 0.81 (0.51–1.11) | 0.99 (0.61–1.36) | 1.36 (0.91–1.81) |
| $\mu_{a_3} * 10^{-5}$ | 0.30 (0.10) | 1.51 (1.27–1.75) | 0.66 (0.43–0.89) | 0.78 (0.49–1.36) | 0.86 (0.59–1.81) |
| $\mu_{b_1}$ | 0.50 (0.10) | 1.20 (0.83–1.57) | 0.54 (0.37–0.71) | 0.86 (0.61–1.11) | 0.58 (0.40–0.77) |
| $b_2 * 10^{-3}$ | 0.30 (0.10) | 7.07 (5.55–8.59) | 3.10 (2.4–3.80) | 5.57 (4.97–6.16) | 7.43 (6.50–8.36) |
| $b_3 * 10^{-4}$ | 0.10 (0.07) | 0.93 (0.03–1.83) | 0.66 (0.02–1.30) | 0.39 (0.01–0.77) | 0.49 (0.01–0.97) |
| $\mu_{c_1}$ | 0.20 (0.05) | 0.11 (0.06–0.16) | 0.13 (0.06–0.20) | 0.05 (0.03–0.07) | 0.30 (0.19–0.42) |
| $\mu_{c_2}$ | 0.12 (0.02) | 0.044 (0.003–0.085) | 0.13 (0.03–0.22) | 0.22 (0.16–0.27) | 0.01 (.0004–0.02) |
| $\mu_{c_3}$ | 0.10 (0.07) | 0.042 (0.002–0.083) | 0.29 (0.18–0.40) | 0.33 (0.25–0.42) | 0.32 (0.23–0.42) |
| $r$ | 1.70 (0.30) | 0.06 (0.01–0.12) | 1.18 (0.83–1.54) | 1.01 (0.83–1.18) | 1.47 (1.10–1.83) |
| $\mu_{y_0} * 10^3$ | | 0.68 (0.68–0.68) | 0.71 (0.35–1.07) | 0.68 (0.36–1.01) | 0.82 (0.46–1.18) |
| $x^*$ | | 0.05 | 1.89 | 1.93 | 2.75 |
| $\sigma_{x_1}$ | | 0.45 | 0.38 | 0.53 | 0.70 |
| $\sigma_{x_2}$ | | 0.22 | 0.40 | 0.33 | 0.22 |
| $\sigma_{x_3}$ | | 0.08 | 0.10 | 0.15 | 0.15 |
| $\sigma_y$ | | 1.00 | 6.02 | 4.50 | 4.01 |

in local conditions, such as variation in the severity of tissue inflammation or control of bacteria at the mucosa level, including changes in the amount of mucus formation and expulsion, had a stronger impact on these fast changes. We also noted that co-infected rabbits sniffled more frequently than *B. bronchiseptica* only animals (Pathak's pers. obs.) and, probably, further contributed to alter the frequency and intensity of these events.

## Sensitivity analysis

To explore in more detail how variation in shedding between types of infection was related to changes in key immune parameters, a sensitivity analysis was performed on model performance (see Materials and methods and Appendix 2). Findings showed that an increase in the rate of bacterial neutralization, $\mu_c$, and, secondly, in the mmune growth rate, $\mu_a$, led to a proportional decrease of the peak of shedding and, to a lesser extent, the time to reach this peak (Appendix 2). Among the three immune variables, the strongest negative impact on shedding was caused by changes in neutrophil ($c_1$ or $a_1$) and IgG ($c_3$ or $a_3$) rates. As expected, an increase in bacterial growth rate, $r$, was associated with a higher peak of shedding and, consequently, a shorter time to reach this peak (*Appendix 1—figures 1 and 2*). Overall, the three co-infections exhibited comparable trends and were more sensitive to immune changes than *B. bronchiseptica* only infected rabbits. The weak response of the latter group was probably caused by the very low bacterial growth rate ($r=0.06$) and thus shedding, suggesting a stronger immune control.

## Discussion

There is increasing evidence that gastrointestinal helminths can impact the severity and time course of respiratory bacterial infections (*Brady et al., 1999*; *Lass et al., 2013*; *Diniz et al., 2010*; *Ezenwa et al., 2010*; *Babu and Nutman, 2016*; *Long et al., 2019*); however, how they affect the dynamics of

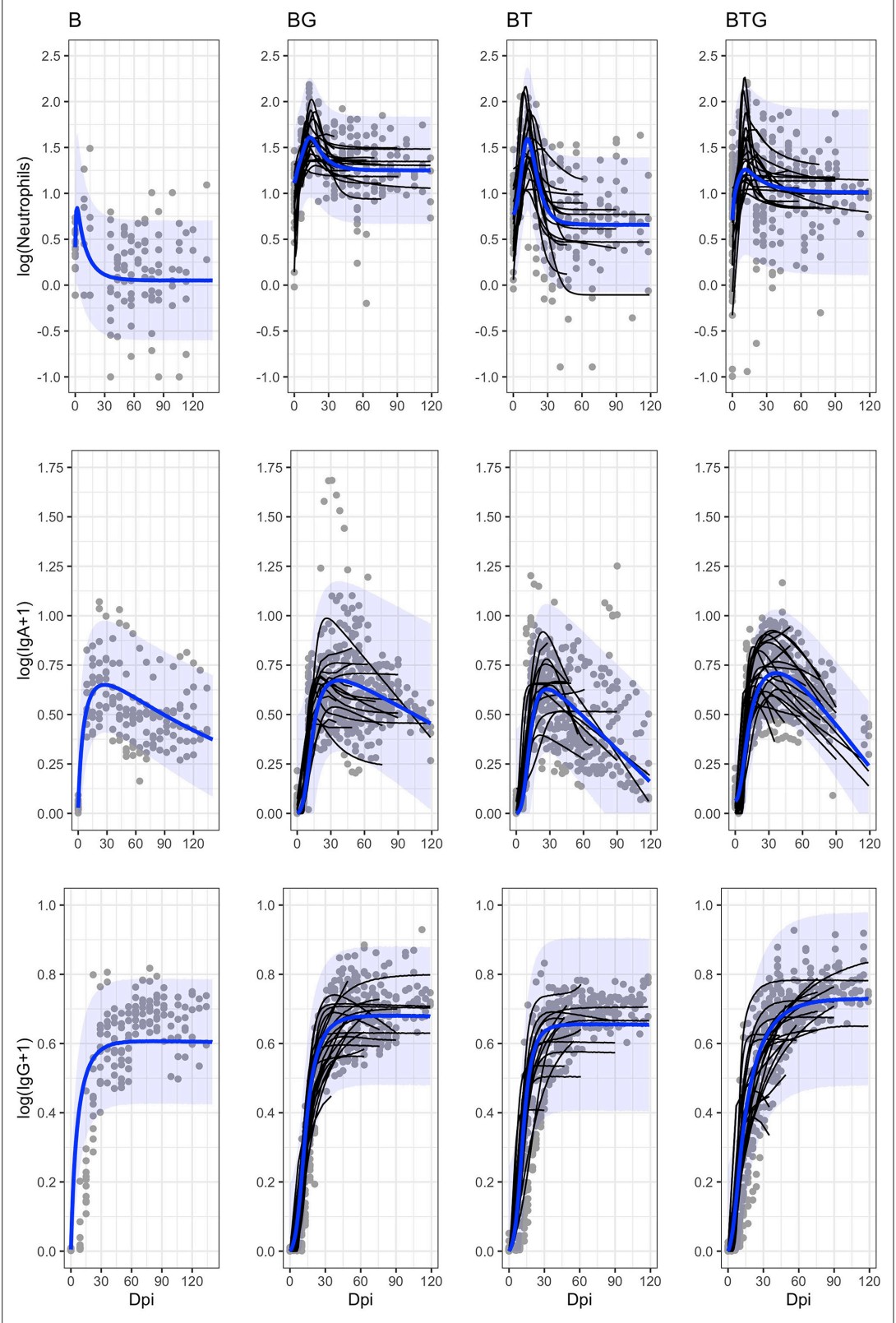

**Figure 5.** Estimated immune responses in the blood by time and type of infection. Empirical longitudinal data (points) with estimated trajectories (black lines) for every individual rabbit, and group means (blue lines) with 95% CIs (blue shadows) are reported. The abrupt interruption of individual data is caused by the removal of rabbits at fixed time points. For the B rabbits we only report the trend at the group level.

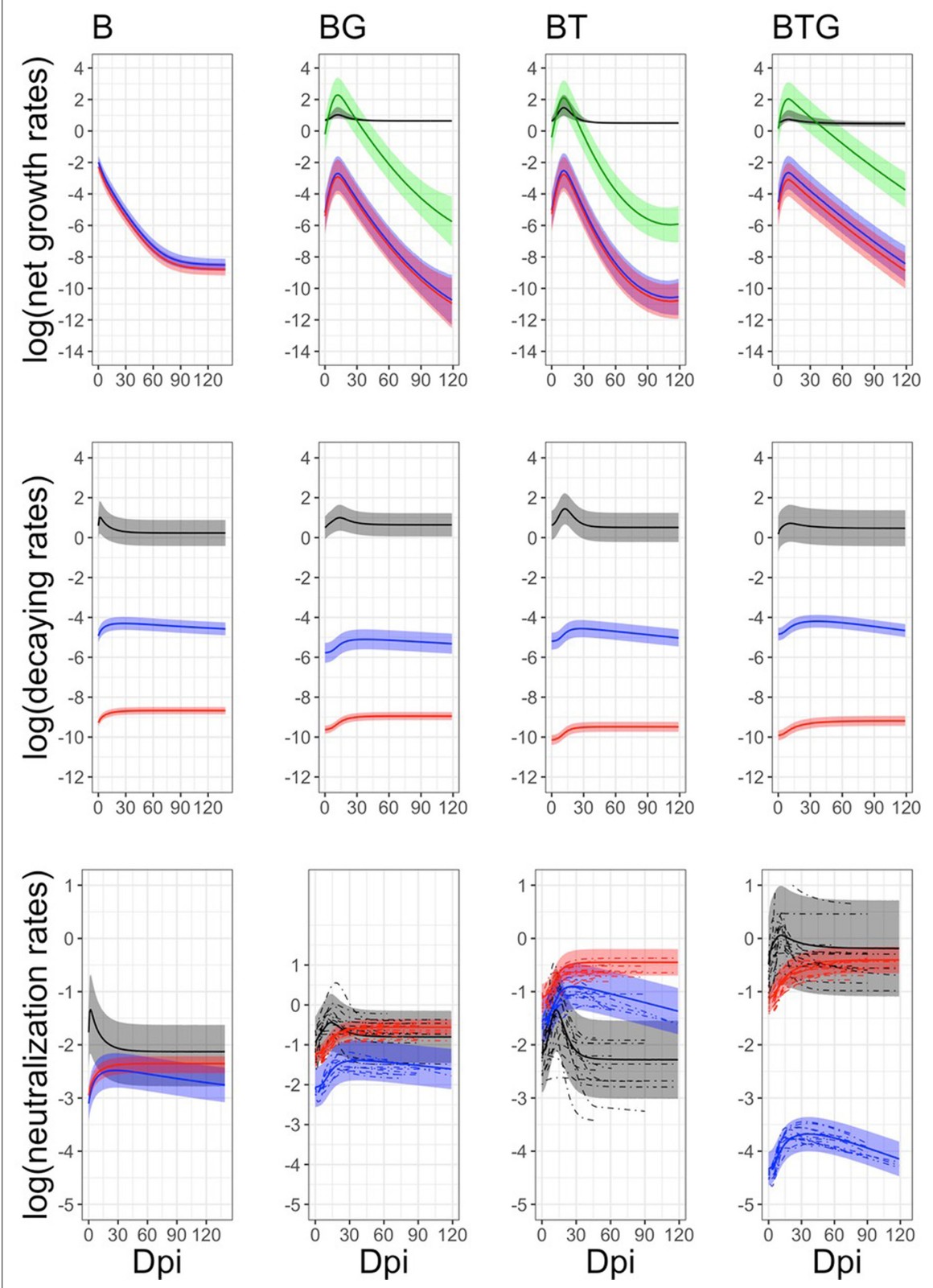

**Figure 6.** Estimated rates of infection in the whole respiratory tract and immune response in the blood by time and type of infection. Means (continuous lines) and 95% CIs (shaded areas) are reported for neutrophils (black), IgA (blue), IgG (red), and *B. bronchiseptica* infection (green). Individual trends (dotted lines) have been included for the neutralization rates of the co-infected groups but not for the other rates, as CIs are very narrow and individual trends are difficult to disentangle. The abrupt interruption of individual trends are caused by the removal of rabbits at fixed time points. Full details on the estimated rates, along with their credible intervals, are available in *Table 2*.

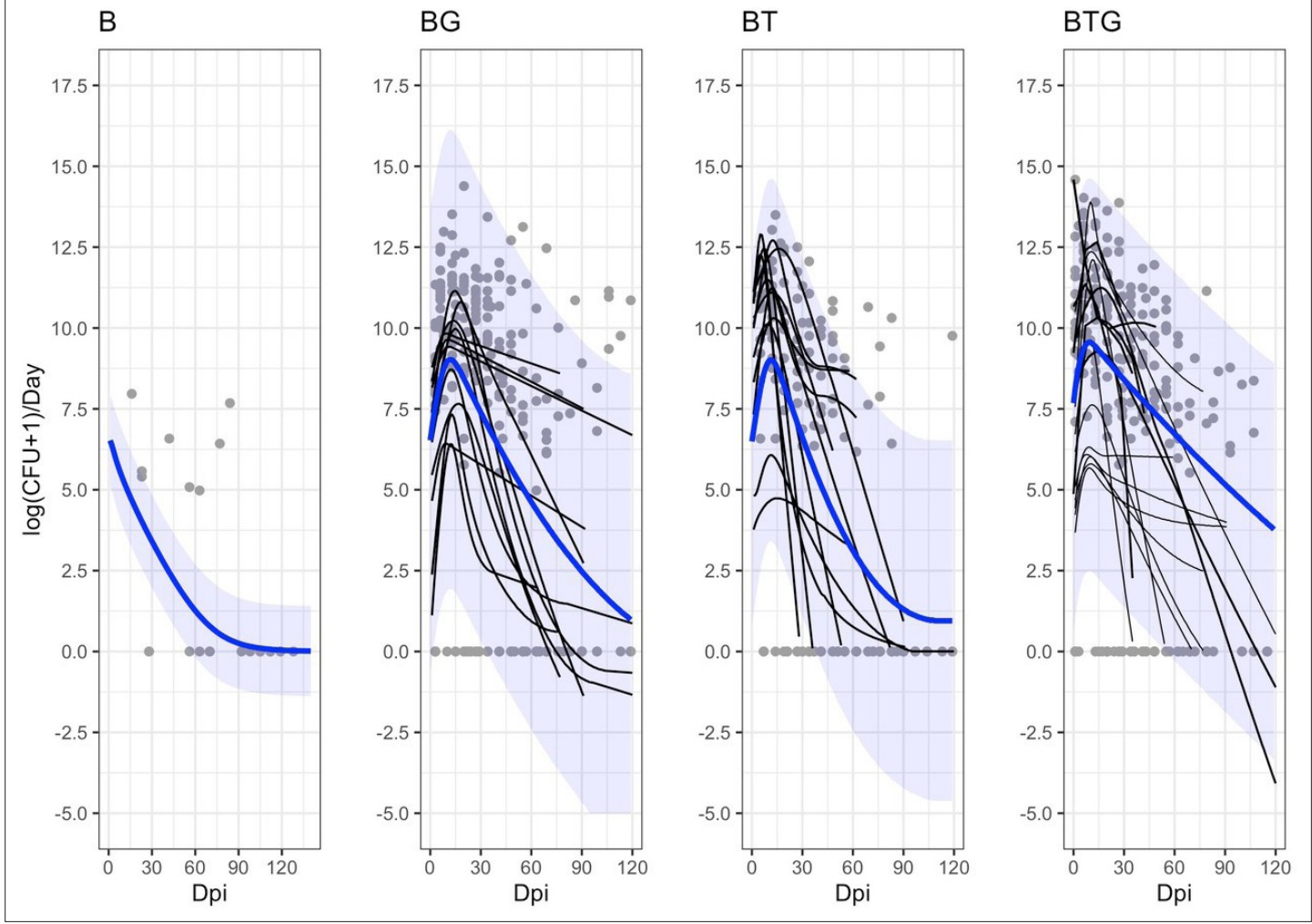

**Figure 7.** Estimated dynamics of *B. bronchiseptica* shedding by time and type of infection. The empirical shedding events (gray points), the estimated individual trajectories (smoothed thin black lines) and the estimated median group trends (blue lines), with the related 95%CIs (blue shadows), are reported. Level of shedding is presented as total daily event to scale up with model dynamics performed at one-day time step. For the B alone rabbits, we only report the group trend. Here, model prediction places the peak at the start of the trial (0 dpi), however, there were no shedding data to train the model during the first ten days and this result warns prudence.

shedding remains to be determined. This is particularly important in regions where chronic helminthiasis co-circulate with respiratory pathogens that are endemic or cause seasonal outbreaks. This is also relevant in areas where antimicrobial resistance is emerging as a threat, such that focusing on the treatment of helminths could be a rapid and effective way to reduce onward bacterial transmission.

We investigated how two gastrointestinal-restricted helminth species, with contrasting infection dynamics, altered the shedding of the highly infectious *B. bronchiseptica* through the modulation of the immune response. Empirical findings showed that the two helminths, taken as a single species or as a pair, significantly increased the level, the variation and the duration of shedding, along with enhancing the frequency of supershedding events. Model estimates suggested that these changes were related to the significant growth of bacteria in the respiratory tract. The developed of an immune response controlled the infection and led to the waning but not the complete clearance of *B. bronchiseptica*. As a consequence of this, the level of shedding decreased with time but rabbits carried on shedding throughout the experiments. We found that changes in the relative contribution of neutrophils, IgA and IgG to bacterial neutralization could explain these dynamics and differences in the pattern of shedding between and within types of infection. However, the rapid changes in the magnitude of these events, commonly observed among rabbits, could not be explained by the relationship between infection and immunity at the scale of analysis selected for this study.

We modeled the dynamics of shedding as representative of the infection in the whole respiratory tract, yet, simulations provided a possible explanation of the contrasting trends in the lungs and nasal cavity. The decline of shedding in the first 90 days post infection could be explained by the immune-mediated clearance of *B. bronchiseptica* from the lungs, previously reported using data from these experiments (*Pathak et al., 2010*; *Pathak et al., 2012*; *Thakar et al., 2012*). In contrast, the reduced but ongoing shedding later in the experiments could be the result of the weak control of bacteria in the nasal cavity (*Pathak et al., 2010*; *Pathak et al., 2012*). We found that neutrophils quickly increased and contributed to bacteria neutralization early on, these were then followed by antibodies that developed more slowly and with an initial delay in the co-infected hosts. The two helminth species appeared not to change the general trend of these variables but the timing and intensity of their relative contribution and, with this, temporal changes in the level of shedding. We previously showed that both helminths elicit a Th-2 immune response (*Murphy et al., 2011*; *Murphy et al., 2013*; *Cattadori et al., 2019*), however, while *T. retortaeformis* was cleared, or significantly reduced from the small intestine, *G. strigosum* persisted with high intensities in the stomach (*Cattadori et al., 2005*; *Cattadori et al., 2008*; *Cattadori et al., 2019*; *Murphy et al., 2011*; *Murphy et al., 2013*; *Mignatti et al., 2016*). It is conceivable to think that the faster decline of shedding in rabbits from the BT than the BG or BTG group could have been caused by the stronger immune control of *B. bronchiseptica*, probably facilitated by the rapid clearance of *T. retortaeformis* (*Thakar et al., 2012*). The stronger immune control could also explain the tendency to the lower shedding and the fewer supershedding events observed. In contrast, the persistence of *G. strigosum* in the stomach probably contributed to delay the decline of shedding and to increase the magnitude of these events, including increasing the frequency of supershedding.

Interestingly, in the BTG triple infection the two helminths appeared to have a positive synergistic effect particularly in delaying the decline of shedding. Given these patterns, and considering that *T. retortaeformis* and *G. strigosum* are commonly found in populations of rabbits, it is not surprising if these helminths play an important role to the rapid transmission and persistence of *B. bronchiseptica* in natural settings. For example, the serological study of a free-living population of rabbits showed that the annual prevalence of *B. bronchiseptica* ranged between 88% and 97% (*Pathak et al., 2010*). Of these rabbits 65% were co-infected with *B. bronchiseptica* and *G. strigosum*, and both bacterial prevalence and helminth intensity increased with rabbit age. Although the study did not examine these patterns in dual or triple infections with *T. retortaeformis*, about 60% of adult rabbits were found to carry BTG infections (unpubl. data), suggesting that rabbits co-infected with helminths represent a large fraction of this host population and, together with the dual infections, are probably responsible for the high prevalence of *B. bronchiseptica* throughout the years.

Recent Boolean modeling of the whole immune network to *B. bronchiseptica* infection indicated that the combined effect of antigen-antibody complex and phagocytosis, activated by neutrophils among others, were important for bacterial removal from the lungs (*Thakar et al., 2012*). The model framework proposed in the current work adds to previous studies by linking a much simplified immune response to the dynamics of shedding through the intensity of infection in the whole respiratory tract. Model estimates showed that the rate of bacterial neutralization by IgA and IgG was stronger in rabbits from the BT group while IgA was particularly low in the BTG rabbits and, secondly, animals from the BG group. These differences could contribute to explain the fast reduction of shedding in the first group and the slower trend in the latter two groups. We found that the much lower rates of antibody neutralization in the *B. bronchiseptica* only rabbits were associated with very low bacterial growth and level of shedding, a pattern consistent with a strong immune control rather than a weak response. The important role of antibodies to *B. bronchiseptica* control was empirically highlighted in the mouse model where serum antibodies (which include IgG isotypes and IgA) cleared *B. bronchiseptica* from lungs and trachea of wild-type and B-cell-deficient mice within 3 days post inoculation (*Kirimanjeswara et al., 2003*). Similarly, IgA was found to be necessary for the reduction of bacterial numbers in the nasal cavity of mice (*Wolfe et al., 2007*).

Our model assumed that neutrophils were stimulated by, and directed against, *B. bronchiseptica* replication in the respiratory tract. The low bacterial shedding in the B group appeared to be associated with the prompt increase of neutrophils and their stronger rate of neutralization, compared to antibodies. In the co-infected rabbits, neutrophils showed to have a more variable role. The neutralization rate was consistently lower in rabbits from the BT group, appeared to have a mixed role with

IgG in the BG group and dominated in the BTG group. In our previous work we found an early peak of neutrophils in the blood of rabbits infected with our two helminths and suggested that this was likely a consequence of gut bacteria infiltrating in the mucosa damaged during helminth establishment in the gastrointestinal tract (*Murphy et al., 2011*; *Murphy et al., 2013*). Although we were not able to disentangle the proportion of neutrophils that were stimulated by and directed against *B. bronchiseptica*, previous work by *Thakar et al., 2012* found no temporal differences in bacterial clearance from the lungs of rabbits from the B and the BT groups, suggesting that there is probably no significant interference from the gut-stimulated neutrophils. The evidence that gut-restricted helminths could enhance the neutrophil response in the respiratory tract was showed in mice co-infected with *Pseudomonas aeruginosa* in the lungs and *Heligmosomoides polygyrus* in the gastrointestinal tract (*Long et al., 2019*). This study found a higher recruitment of neutrophils in the lungs together with an increase of CD4 +T cells and Th2 cytokine expression. Interestingly, survival rate was improved in co-infected mice than mice with bacteria alone, suggesting that neutrophils, rather than a Th2 response, dominated in the lungs. Work by *Rolin et al., 2014* showed that TLR4-mutant mice infected with *B. bronchiseptica* poorly controlled bacterial growth and shedding. Animals exhibited heavy neutrophil infiltration in the lungs, however, the depletion of neutrophils did not affect the level of infection but decreased individual shedding. Taken together, we propose that the possible interference with a Th2 immune signal stimulated by *T. retortaeformis* and *G. strigosum* contributed to increase the level and duration of *B. bronchiseptica* shedding, although this anti-inflammatory response was not sufficiently strong to prevent bacteria clearance from the lower respiratory tract. We also note that while we focused on neutrophils and antibodies, this interference most likely involved other immune variables important in the network of immune reactions (*Thakar et al., 2012*).

A common feature among rabbits, irrespective of the infection group they belong to, was the rapid fluctuation in the level of bacteria shed over time, including the unpredicted supershedding events. We could not explain these rapid changes based on our scale of analysis. Most likely, other processes generated the exponential bursts and timing of (super)shedding observed. These could include, but are not limited to, immune-mediated control of bacteria turnover in the mucosa or stochastic events during bacteria release, such as random mucus discharge, and also variation in host behavior. Our finding is consistent with other systems where fast, intermittent shedding was reported both from single (*Hadinoto et al., 2009*; *Schiffer et al., 2009*; *Rolin et al., 2014*; *Spencer et al., 2015*; *Slater et al., 2016*) and co-infected hosts (*Byrne et al., 2019*; *Kao et al., 2007*). A compelling study by *Hadinoto et al., 2009* on Epstein-Barr virus shedding in healthy carriers showed that the virus was rapidly and continuously shed in the saliva but the process of virus production and control was regulated by multiple factors, including the immune response. Importantly, they showed that variation in the level and frequency of shedding was a process that occurred at the individual level over time. Our study further supports this finding by showing variation in *B. bronchiseptica* shedding at three levels: between types of infection, between hosts with each infection type, and within each host over time. Our model framework, and the scale of our analysis, was able to describe the first two types of variation but prevented us from capturing the local mechanism responsible for the rapid changes observed at the individual level. More work at the local tissue level is needed to explain these patterns, including the processes generating supershedding events and the relative contribution of the lower and upper respiratory tract.

As illustrated by our findings, the definition of supershedder can become problematic when hosts are infected with more than one parasite/pathogen. Our analysis showed that a cut-off value based on single infection can lead to the overestimation of supershedding in co-infected rabbits. However, by pulling the data together we can calculate a common percentile threshold and provide a more accurate estimation. We used two closely related helminth species and if we consider the 99th percentile threshold, we found that between 18% and 32% of the co-infected rabbits generated at least one supershedding event, which greatly contrast with the lack of cases from the *B. bronchiseptica* only rabbits. These are significant percentages in our co-infected groups, especially if we consider that about 30% of the shedding events were null. Therefore, we can speculate that given the magnitude and frequency of supershedding, the risk of onwards transmission following a contact is likely to non-linearly increase for the co-infected hosts.

This study adds novel insights into the role of gastrointestinal helminths to the dynamics of *B. bronchiseptica* shedding. We used a parsimonious mechanism of immune regulation based on previous

work by others and ourselves. While we focused on antibodies and neutrophils other immune variables could have contributed to the dynamics of shedding observed. Our model framework can be adapted to include the impact of other factors, such as components of the innate immune response or hierarchical relationships between immune variables. To reduce model complexity, we did not explicitly quantify the dynamics of infection of the two helminths although this can be explored in future work, including the role of *B. bronchiseptica* to the dynamics of shedding by the two helminths.

Given that one quarter of the global human population is infected with helminths and considering that respiratory infections are among the top 10 causes of death by infectious diseases worldwide, understanding the modulatory role of helminth species to respiratory infections is important for developing treatments targeted to specific co-infection settings. The ability to detect and, ideally, control the high shedders and/or supershedders is also critical for reducing the risk of disease outbreak and spread, and should not be overlooked any longer.

## Materials and methods

### Ethic statement

Animals were housed in individual cages with food and water ad libitum and a 12 hr day/night cycle, in compliance with Animal Welfare Act regulations as well as the Guide for the Care and Use of Laboratory Animals. All animal procedures, including infections with *B. bronchiseptica* and the two helminth species, weekly blood collection and pathogen/parasite sampling at fixed time, were approved by the Institutional Animal Care and Use Committee of The Pennsylvania State University (IACUC 26082). All animal work complied with guidelines as reported in the Guide for the Care and Use of Laboratory Animals, 8th ed. National Research Council of the National Academies, National Academies Press Washington DC.

### Bacteria strain and culture

We used *B. bronchiseptica* strain RB50 for all experiments. Bacteria were grown on Bordet-Gengou (BG) agar supplemented with 10% defibrinated sheep blood and streptomycin (20 µg/ml). The inoculum was prepared by growing the bacteria in Stainer-Scholte (SS) liquid culture medium at 37 °C overnight. For the infection, bacteria were re-suspended in sterile phosphate-buffered saline (PBS) at a density of $5 * 10^4$ CFU/ml, which was confirmed by plating serial dilutions of the inoculum on BG blood agar plates in triplicate (*Pathak et al., 2010*).

### Laboratory infections

*B. bronchiseptica* single infection (B) and dual infections with either *T. retortaeformis* (BT) or *G. strigosum* (BG) are described in detail in *Pathak et al., 2010*; *Pathak et al., 2012* and *Thakar et al., 2012*. The triple infection (BTG) followed the same experimental design and laboratory procedures of the dual infections. Here, we describe the general design of the experiments. New Zealand White, two months old, male rabbits (Harlan, USA), were challenged with *B. bronchiseptica* and two helminth species as follow: i- *B. bronchiseptica* (B) single infection: 32 infected and 16 controls, ii- *B. bronchiseptica-Graphidium strigosum* (BG): 31 infected and 16 controls, iii- *B. bronchiseptica-T. retortaeformis* (BT): 32 infected and 16 controls, and iv- the three agents together (BTG): 32 infected and 16 controls. Infection was performed by pipetting in each nare $2.5 * 10^4$ CFU/ml of bacteria diluted in 0.5 ml of PBS. For the co-infections animals also received, simultaneously by gavage, a single inoculum of water (5 ml) with either 5500 *T. retortaeformis* or 650 *G. strigosum* third stage infective larvae, or both. Helminth doses followed natural infections (*Cattadori et al., 2005*; *Cattadori et al., 2008*). Control animals were sham inoculated with 1 ml of sterile PBS in the nares and gavaged with 5 ml of water. The dynamics of infection and related immune responses were then followed for 120 days (150 days for B) by sacrificing four infected and two control animals at fixed days post infection, as follow, B: 3, 7, 15, 30, 59, 90, 120, 150 days; BG: 7, 14, 30, 44, 62, 76, 90, 120 days; BT: 5, 8, 15, 31, 47, 61, 91, 120 days; BTG 7, 14, 30, 45, 60, 75, 90, 120 days. This sampling design follows important time points in the life cycle of the three infections (*Murphy et al., 2011*; *Murphy et al., 2013*; *Pathak et al., 2010*), a shift of 1 or 2 days between infections was necessary a few times to adjust with our laboratory activities.

## Bacteria shed enumeration

At the start of each infection, a subset of infected rabbits was randomly selected (i.e. B=16, BG = 23, BT = 20 and BTG = 24) to quantify the level of bacteria shed by contact with a BG-blood agar petri dish over time (*Pathak et al., 2010*). Shedding by contact with a surface mimics the natural transmission of a respiratory pathogen without disruption of the bacteria population through swabbing. Shedding was assessed once a week for B single infection and 2 (BT) or 3 (BG and BTG) times a week for the co-infected rabbits. The use of a different number of hosts and frequency of sampling was determined by logistical constraints (i.e. personnel availability). For the *B. bronchiseptica* group the use of a different substrate (plastic balls) the first 10 days post-infection was not successful and we missed to record the dynamics of shedding in those early days. At every sampling point, rabbits were allowed to interact with the petri dish by direct oral-nasal contact and for a maximum of 10 min. Plates were removed earlier if animals chewed the plastic or the agar, and the duration of each interaction was recorded. Plates were then incubated at 37 °C for 48 hr and colonies counted and scaled to the interaction time (CFU/s). If there was an interaction but plates resulted negative, shedding was considered to be null while the lack of interaction (e.g. animals were not interested in the plate) was recorded as a missing point.

## Systemic immune response

As representative of the immune response to *B. bronchiseptica*, we selected neutrophils and species-specific antibodies IgA and IgG. Previous laboratory experiments and related modeling studies suggested that these three variables contribute to clear *B. bronchiseptica* from the lungs and to reduce the colonization in the nasal cavity (*Orndorff et al., 1999*; *Kirimanjeswara et al., 2003*; *Wolfe et al., 2007*; *Thakar et al., 2007*; *Thakar et al., 2012*). For example, results from modeling the immune network in the lungs of mice and rabbits showed that the lack of antibody production, by B cells deletion, prevented bacterial clearance (*Thakar et al., 2007*; *Thakar et al., 2012*). Similarly, peripheral neutrophils recruited via pro-inflammatory cytokines contributed to the activation of phagocytic cells and bacterial neutralization (*Thakar et al., 2007*; *Thakar et al., 2012*). Consistent with these findings, experimental studies showed that adoptive transfer of serum antibodies in mice led to the removal of *B. bronchiseptica* by day 3 post inoculation *Kirimanjeswara et al., 2003*. Methodologies to quantify neutrophils, IgA and IgG are described in *Pathak et al., 2010*; *Pathak et al., 2012* and *Thakar et al., 2012*. Briefly, for every rabbit blood was collected once a week for neutrophils and twice a week for antibodies. Neutrophil concentration was measured using whole blood (0.2 ml) stored in EDTA (Sartorius, Germany) and analyzed using Hemavet-3 hematology system (Drew Scientific, USA). Species-specific IgA and IgG were estimated from blood serum using *B. bronchiseptica* as a source of antigen and ELISA (*Pathak et al., 2010*). Measurements were performed in duplicates with all plates having high, low and background controls. Values were expressed as immunosorbent Optical Densities (OD) and then standardized into Optical Density index (*Murphy et al., 2011*). Plate preparation and dilutions, including the preparation of high (strongly reacting animals) and low (non-reacting animals from prior to the infection) pools and checkerboard titrations, are detailed elsewhere (*Pathak et al., 2010*; *Pathak et al., 2012*; *Thakar et al., 2012*). For the triple infection, antibody quantification and dilutions followed *Pathak et al., 2010*. The baseline profile of the three immune variables were available from blood data collected from every rabbit the week before the infection; if individual data were missing we used the average value from control rabbits. Neutrophils rapidly returned to baseline levels following the initial infection and the equilibrium value was available as the average from control rabbits sampled throughout the experiments, specifically cells/ml B: 1.323, BG: 2.730, BT: 1.819 and BTG: 2.289.

## Model framework

The within-host mechanisms that affect the dynamics of *B. bronchiseptica* shedding were examined in two steps. First, we developed a dynamical model that describes the dynamics of neutrophils, specific IgA and IgG, and their interaction with the bacterial infection in the whole respiratory tract. Second, a Bayesian approach was then used to link this dynamical model to the empirical longitudinal data by: i- fitting the model to the three immune variables of every rabbit and ii- fitting the intensity of *B. bronchiseptica* infection, estimated from the dynamical model, to the experimental shedding data. Below we describe the dynamical model and in the next sections we report on the observation model

and model fitting to the empirical data, including how parameter calibration was performed. For completeness, we also include two additional sections: i- model validation that explores the accuracy of our model and ii-model selection where we compare three different model formulations.

## Dynamical model

Different modeling approaches have been applied to study the within-host dynamics of infection of *Bordetella* species and related immune response. Work by *de Graaf et al., 2014* on *B. pertussis* followed a phenomenological approach based on a Bayesian hierarchical framework that described the rise and decline of IgG data as a response to the infection. In contrast, previous studies by colleagues and us focused on a network-based dynamical model wherein relationships between components of the immune system against *B. bronchiseptica* or *B. pertussis* infection were examined using experimental data and Boolean transfer functions (*Thakar et al., 2007*; *Thakar et al., 2012*). This approach allowed us to follow the complexity of the immune system both at the systemic and localized level of infection and the consequences on bacterial control. In the current study, we developed a deterministic dynamical model that explored the relationship between bacterial shedding and host immunity using an individual based Bayesian approach and laboratory data. Here, we simplified the immune response down to three variables and examined their direct effect on the dynamics of infection and related shedding. The time-dependent interactions between immune variables and *B. bronchiseptica* infection are described by the following system of ordinary differential equations:

$$\frac{dx_{1,i}}{dt} = a_{1,i}y_i - b_{1,i}(x_{1,i} - x_1^*) \tag{1}$$

$$\frac{dx_{2,i}}{dt} = a_{2,i}y_i - b_2 x_{2,i} \tag{2}$$

$$\frac{dx_{3,i}}{dt} = a_{3,i}y_i - b_3 x_{3,i} \tag{3}$$

$$\frac{dy_i}{dt} = (r_i - c_{1,i}x_{1,i} - c_2 x_{2,i} - c_3 x_{3,i}) * y_i \tag{4}$$

where $x_{1,i}(t)$, $x_{2,i}(t)$ and $x_{3,i}(t)$ are neutrophils, IgA, and IgG, respectively, of rabbit $i^{th}$ at time $t$. $y_i(t)$ is the intensity of bacteria in the respiratory tract of this same rabbit at time $t$. The parameters $a_{1,i}$, $b_{1,i}$ and $c_{1,i}$ describe the rate of growth, decay and bacteria clearance, respectively, for neutrophils in rabbit $i^{th}$. Similarly, $a_{2,i}, b_2$ and $c_{2,i}$ represent the per capita rates of growth, decay and bacterial clearance for IgA, while $a_{3,i}$, $b_3$ and $c_{3,i}$ are the rates representing IgG (*Table 1*). The baseline immune conditions of the host before the infection are $x_{1,i}(0)$, $x_{2,i}(0)$ and $x_{3,i}(0)$, while $x_1^*$ describes the equilibrium level of neutrophils post infection; this was not included for IgA or IgG since the time to reach equilibrium extended beyond the course of the experiments. The parameter $r_i$ represents the per capita bacterial replication rate. The full description of model parameters and variables is reported in *Table 1*. In our model formulation, we made the parsimonious assumption that the activation and response of neutrophils and specific IgA and IgG proportionately increase with the intensity of infection, $y_i(t)$, in the whole respiratory tract. Similarly, bacterial neutralization occurs through the additive effect of these three immune variables. These direct interactions are a large simplification of a more complex immune process (*Thakar et al., 2007*; *Thakar et al., 2012*), and follow a classical Lotka-Volterra type of relationship commonly used to describe within-host processes of infection (*Mohtashemi and Levins, 2001*; *Pugliese and Gandolfi, 2008*; *Fenton and Perkins, 2010*; *de Graaf et al., 2014*; *Vanalli et al., 2020*). To reduce model complexity we assumed that the two helminths affect *B. bronchiseptica* infection, and thus shedding, by altering the magnitude and time course of neutrophil, IgA and IgG responses.

## Observation model

We applied the dynamical model to every rabbit (except for the *B. bronchiseptica* group). Individual parameters could vary among hosts as independent samples from a joint log-normal distribution, while parameters that were shared among rabbits (see below) were kept the same for that group. This hierarchical set-up allowed us to have individual responses that varied in term of amplitude, time to peak and decay rate, while keeping these trends from deviating too much from each other. For a given rabbit $i^{th}$ at time $t$, every empirical immune variable, log-transformed, was assumed to follow a normal distribution with mean, $\mu$, and variance, $\sigma$, as:

$$\log(Ne_{i,t}) \sim \mathcal{N}(\log(x_{1,i}(t)), \sigma_{x_1}) \tag{5}$$

$$\log(I_{A,i,t}) \sim \mathcal{N}(\log(x_{2,i}(t)), \sigma_{x_2}) \tag{6}$$

$$\log(I_{G,i,t}) \sim \mathcal{N}(\log(x_{3,i}(t)), \sigma_{x_3}) \tag{7}$$

The empirical amount of *B. bronchiseptica* shed by rabbit $i^{th}$ at time *t*, is directly proportional to the level of infection $y_i(t)$ and is assumed to be representative of the intensity of infection in the whole respiratory tract. The probability of having a shedding event is independent of time since inoculation, in that shedding can occur anytime during the experiment and anytime during the interaction with the petri dish. Shedding was then related to the dynamics of infection via a zero-inflated log-normal relationship, to account for the high fraction of events in which rabbits did not shed (i.e. null shedding) despite been infected (*Figure 3*), as:

$$\log(S_{i,t}) \sim w_{0,i} + (1 - w_{0,i}) * \mathcal{N}(\log(y_i(t)), \sigma_y) \tag{8}$$

where $w_{0,i}$ is the fraction of events when a given host did not shed. For consistency with the immune data, collected on a daily scale, the estimated shedding was then scaled up and quantified as total amount of bacteria shed by a rabbit in a day.

## Parameter calibration and model fitting

The dynamical model described in *equations (1)-(4)* includes parameters that are shared among rabbits within the same group, specifically:

$$\boldsymbol{\theta_1} = \{\mu_{a_1}, \sigma_{a_1}, \mu_{b_1}, \sigma_{b_1}, \mu_{c_1}, \sigma_{c_1}, \mu_{a_2}, \sigma_{a_2}, b_2, \mu_{c_2}, \sigma_{c_2},$$
$$\mu_{a_3}, \sigma_{a_3}, b_3, \mu_{c_3}, \sigma_{c_3}, \mu_r, \sigma_r\} \tag{9}$$

where $\mu$ and $\sigma$ are the means and standard deviations, respectively, of the corresponding normal distributions of the parameters *a*, *b*, *c* and *r* for each group. Likewise, the set of parameters $\boldsymbol{\theta_{2,i}}$ represents the estimates of every individual rabbit within a group, as:

$$\boldsymbol{\theta_{2,i}} = \{a_{1,i}, b_{1,i}, c_{1,i}, a_{2,i}, c_{2,i}, a_{3,i}, c_{3,i}, r_i, w_{0,i}\}_{(i \in 1,\dots,N)} \tag{10}$$

where N is the number of infected rabbits in each group. The resulting hyper prior distribution for rabbit $i^{th}$ is:

$$p(\boldsymbol{\theta_{2,i}}) = \phi(\log(a_{1,i})|\mu_{a_1}, \sigma_{a_1}) * \phi(\log(b_{1,i})|\mu_{b_1}, \sigma_{b_1}) * \phi(\log(c_{1,i})|\mu_{c_1}, \sigma_{c_1}) *$$
$$\phi(\log(a_{2,i})|\mu_{a_2}, \sigma_{a_2}) * \phi(\log(c_{2,i})|\mu_{c_2}, \sigma_{c_2}) *$$
$$\phi(\log(a_{3,i})|\mu_{a_3}, \sigma_{a_3}) * \phi(\log(c_{3,i})|\mu_{c_3}, \sigma_{c_3}) * \phi(\log(r_i)|\mu_r, \sigma_r) \tag{11}$$

The resulting joint-likelihood function for the observed longitudinal time-series of all infected rabbits is:

$$\mathcal{L}(\boldsymbol{\theta_1}, \boldsymbol{\theta_2}) \propto p(\boldsymbol{\theta_1}) * \prod_{i=1}^{N} p(\boldsymbol{\theta_{2,i}})$$
$$* \prod_{i=1}^{N} \prod_{j=1}^{n_i} \phi(\log(Ne_{i,j})|\log(x_{1,i,j}), \sigma_{x_1})$$
$$* \prod_{i=1}^{N} \prod_{k=1}^{n_{1,i}} \phi(\log(I_{A,i,k})|\log(x_{2,i,k}), \sigma_{x_2})$$
$$* \prod_{i=1}^{N} \prod_{l=1}^{n_{2,i}} \phi(\log(I_{G,i,l})|\log(x_{3,i,l}), \sigma_{x_3})$$
$$* \prod_{i=1}^{N} \prod_{h=1}^{n_{S,i}} \phi(\log(S_{i,h})|\log(y_{i,h}), \sigma_y) \tag{12}$$

where $n_i$, $n_{1,i}$, $n_{2,i}$, and $n_{S,i}$ are the number of measurements for neutrophils, IgA, IgG and bacteria shed, respectively, for rabbit $i^{th}$. The four subscripts *j, k, l, h* are the time index for neutrophils, IgA, IgG and bacteria shed, respectively, at the specific time points for rabbit $i^{th}$. The notation $\phi(x|\mu, \sigma^2)$ represents the normal probability density of *x* with mean $\mu$ and variance $\sigma^2$. Since there is no knowledge on the values of $\boldsymbol{\theta_1}$ we used normal priors as reported in *Table 2*.

Model fitting, and the generation of posterior distributions for the parameters estimated, was performed using Hamiltonian Monte Carlo (HMC) algorithm implemented in Stan package version 2.18.0. Briefly, the algorithm was implemented as follow:

1. Start sampling from the joint normal prior distributions for $\theta$, here referring to all parameters included in $\theta_1$ and $\theta_2$, (the initial values $\theta^0$ could be user-specific or specified by Stan). Set the initial value of $\theta$ as the current $\theta^*$.

2. Evaluate momentum vector for the current $\theta^*$ (Note that in HMC, the momentum vector $\mathcal{L}(\theta)$ is described by its 'kinetic' and 'potential' energies).

3. The HMC proposes a new $\theta^*$ by sampling from the posterior distributions of $p(\theta|y)$ with a given stepsize which is optimized by the algorithm.

4. To account for numerical errors during integration, a Metropolis acceptance step is applied. If the HMC proposed momentum vector has a higher probability than the previous parameter values, the proposed values will be updated to the current values and used to initialize the next iteration. If the Metropolis acceptant rejects the proposed values, the previous values are returned and used to initialize the next iteration. Repeat the following steps 2–4 until the maximum number of iterations is reached. This procedure is for a single chain, there are four parallel chains, each with 100,000 iterations including 40,000 burn-in iterations. This procedure was repeated for every rabbit in each of the four types of infection.

5. The generated sampling distribution represents the posterior distribution for each parameter, once the warm-up iterations have been removed. The posterior distribution is now used to quantify the statistics of interest (e.g. mean, median, 95% CI) for each group.

Model simulations were performed at one-day time step increment and parameters were scaled to account for differences in their relative magnitude. To provide meaningful results and to avoid problems with model convergence, rabbit time series with three, or less, shedding points larger than zero were excluded. This led us to used: 17 BT, 22 BG, 23 BTG, and 7 B rabbits. Given the small number of rabbits, and individual time series with few shedding events larger than zero, the modeling of the *B. bronchiseptica* only rabbits was carried out only at the group level.

To ensure proper model convergence, we assessed: i- the scale reduction factor $\hat{R}$ on split chains, to confirm that the value is close to 1 for each parameter, a $\hat{R} > 1.1$ is usually an indicator of a fit problem, and ii- the crude measure of the effective sample size $n_{eff}$, this value should be close to 1 for a good parameter estimate. Moreover, the Rstan package allows us to set the maximum trajectory length to avoid infinite loops that can occur for non-identified models. If a high proportion of the interactions saturate the maximum threshold, the model is not effectively sampled. Finally, we checked the number of divergences, the trace plot of the Markov Chain time-series and the plot of the priors and posteriors for each parameter. All parameter estimates were diagnosed with the above criteria to ensure that the chains were well-mixed with no divergent transitions post-burning (Appendix 3).

## Sensitivity analysis

To examine how key immune parameters affected shedding, a sensitivity analysis was performed (Appendix 2). As representative of shedding, we considered the peak of shedding and the time to reach the peak, including bacteria growth rate $r$, while for the immune variables we selected the rate of bacterial neutralization $\mu_c$ and the growth rate $\mu_a$. Analyses were carried out using mean values for each type of infection. Specifically, for either $c$ or $a$ the estimated optimal value presented in *Table 2* was changed by an incremental percentage, while keeping all the other parameters fixed, and the consequent changes in the amplitude and timing of the shedding peak was quantified. The same exercise was repeated using incremental changes of $r$; 1000 simulations were performed for each parameter at each step.

## Model validation

As part of our technical check of model performance, to investigate whether the Bayesian model could recover the starting parameter values we performed a simulation-based validation using parameters at the group level. Briefly, we randomly drew $n$ sets of parameters $\theta$ from our prior distributions (n=1,000 iterations) and generated a $y_i$ for each parameter set of $\theta_i$ from our dynamical model. We then fitted the simulated dataset to our Bayesian framework to confirm that the posterior distributions

included the known sets of parameters. The model showed a good ability to recover the original parameters (Appendix 4).

## Model selection

To examine the effect of different immune responses on the pattern of shedding, three model formulations were compared: i- neutrophils, IgA and IgG (full model), ii: neutrophils and specific IgA (reduced model A) and iii: neutrophils and specific IgG (reduced model G), where neutrophils were kept as the fundamental variable. To simplify the workflow, model selection was performed using data from each group in the form of time series of geometric means for every variable for each of the four infections, group level parameters were also used. The best model was selected based on the best compromise between goodness of fit and parsimony, according to the Bayesian information criterion (BIC). Results showed that the full model was statistically better in capturing the dynamics of *B. bronchiseptica* shedding, than the reduced formulations (Appendix 5). This model was then used to perform simulations both at the level of individual rabbit and infection group, with the exception of the *B. bronchiseptica* only rabbits that were examined at the group level.

## Acknowledgements

The authors thank Kathleen Creppage and Chad Pelensky for their research assistance during animal work and sample processing. This study, IMC and AKP were supported by Human Frontier Science Program (RGP0020/2007 C) and National Science Foundation-DEB (1145697). The funders had no role in study design, data collection and analysis, manuscript preparation and decision to publish.

## Additional information

### Funding

| Funder | Grant reference number | Author |
|---|---|---|
| Human Frontier Science Program | RGP0020/2007-C | Ashutosh K Pathak |
| National Science Foundation | 1145697 | Ashutosh K Pathak |

The funders had no role in study design, data collection and interpretation, or the decision to submit the work for publication.

### Author contributions

Nhat TD Nguyen, Formal analysis, Validation, Methodology, Writing – original draft, Writing – review and editing; Ashutosh K Pathak, Data curation, Methodology, Writing – review and editing, Animal work, laboratory analysis; Isabella M Cattadori, Conceptualization, Funding acquisition, Investigation, Writing – original draft, Project administration, Writing – review and editing

### Author ORCIDs

Isabella M Cattadori (iD) http://orcid.org/0000-0001-6618-316X

### Ethics

Animals were housed in individual cages with food and water ad libitum and a 12hr day/night cycle, in compliance with Animal Welfare Act regulations as well as the Guide for the Care and Use of Laboratory Animals. All animal procedures, including infections with the bacterium and the two helminth species, weekly blood collection and pathogen/parasite sampling at fixed time, were approved by the Institutional Animal Care and Use Committee of The Pennsylvania State University (IACUC 26082). All animal work complied with guidelines as reported in the Guide for the Care and Use of Laboratory Animals. 8th ed. National Research Council of the National Academies, National Academies Press Washington DC.

### Decision letter and Author response

Decision letter https://doi.org/10.7554/eLife.70347.sa1

Author response https://doi.org/10.7554/eLife.70347.sa2

## Additional files

### Supplementary files
• Transparent reporting form

### Data availability

All empirical data used in this study are available at https://doi.org/10.5061/dryad.g79cnp5sx. File titles: Nguyen-Pathak-Cattadori eLife Bordetella shedding.csv; Nguyen-Pathak-Cattadori eLife neutrophils.csv; Nguyen-Pathak-Cattadori eLife IgA ODI.csv; Nguyen-Pathak-Cattadori eLife IgG ODI.csv; README.doc.

The following dataset was generated:

| Author(s) | Year | Dataset title | Dataset URL | Database and Identifier |
|---|---|---|---|---|
| Nguyen N, Pathak A, Cattadori IM | 2022 | Gastrointestinal Helminths Increase Bordetella bronchiseptica Shedding and Host Variation in Supershedding | https://dx.doi.org/10.5061/dryad.g79cnp5sx | Dryad Digital Repository, 10.5061/dryad.g79cnp5sx |

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

## Appendix 1

### Parameter estimates from posterior distributions

Averages of posterior parameter estimations at the individual and group level were examined for the three co-infection groups (BG, BT and BTG); *B. bronchiseptica* alone rabbits were excluded since simulations were based on the whole group. The decay rates of IgA and IgG, $b_2$ and $b_3$, respectively, were assumed to be the same for rabbits of each group and were not reported. Results show clear rabbit variation in *B. bronchiseptica* dynamics of infection and immune responses, both within and between groups (*Appendix 1—figures 1–3*).

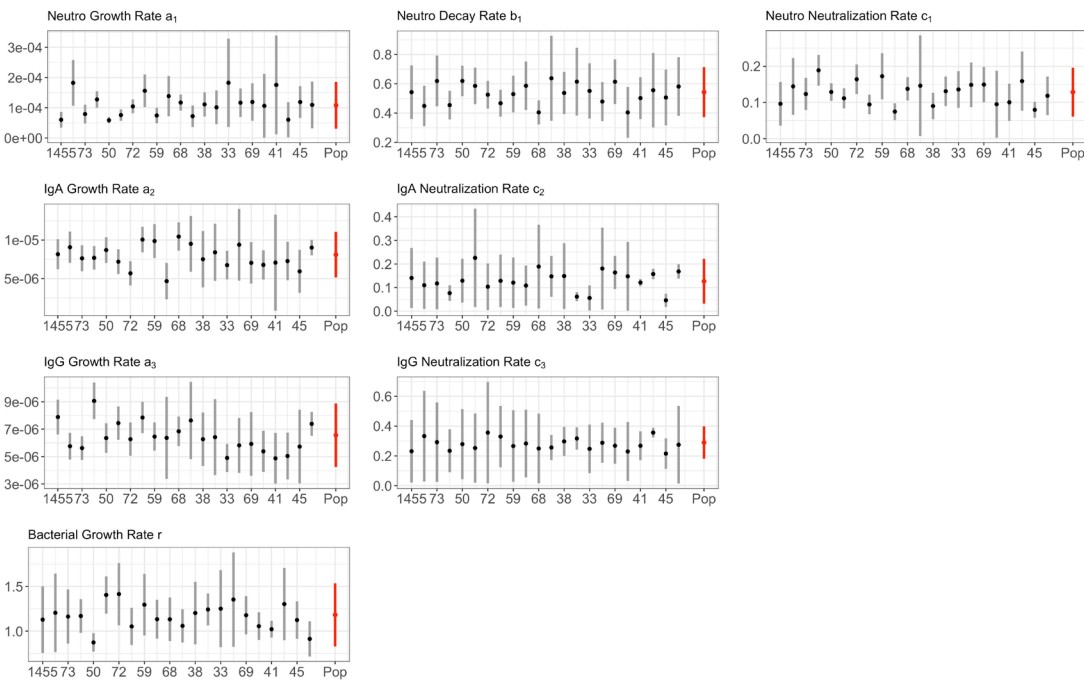

**Appendix 1—figure 1.** Parameter estimates from posterior distribution for BG. Individual average (black points) with 95% CIs (black segments), and group average (named 'Pop', red points) with 95% CIs (red segments), are reported. X-axis lists the rabbit's ID every 5 animals using the same left-to-right ID order, and host sampling scheme, as detailed in *Figure 1* of the main text.

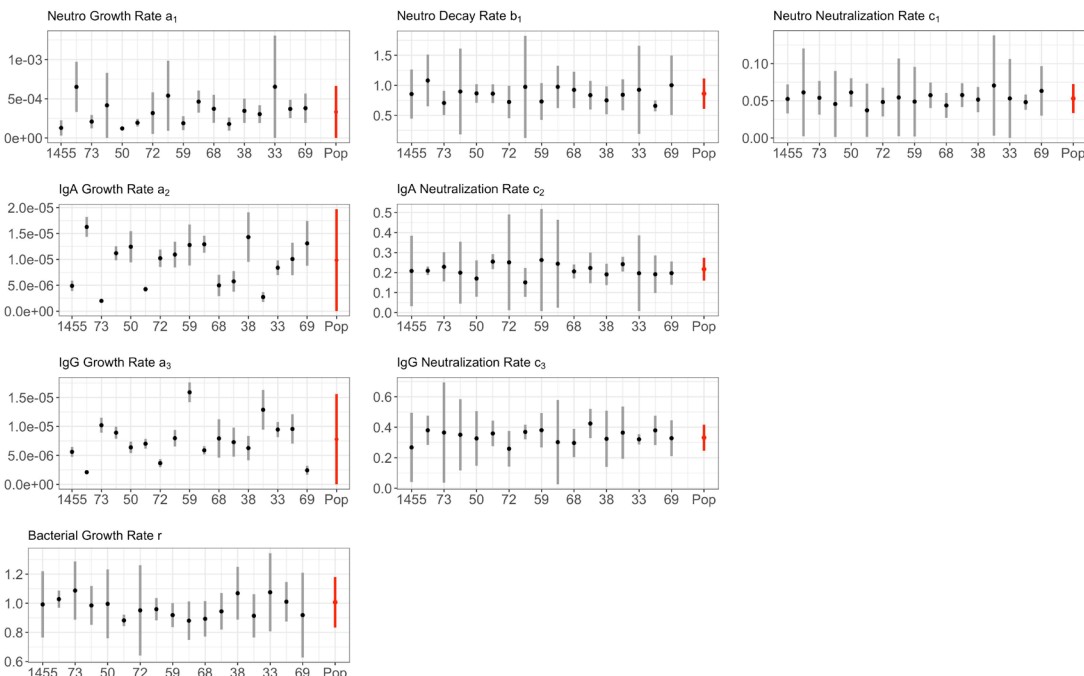

**Appendix 1—figure 2.** Parameter estimates from posterior distribution for BT. Full details in *Appendix 1—figure 1*.

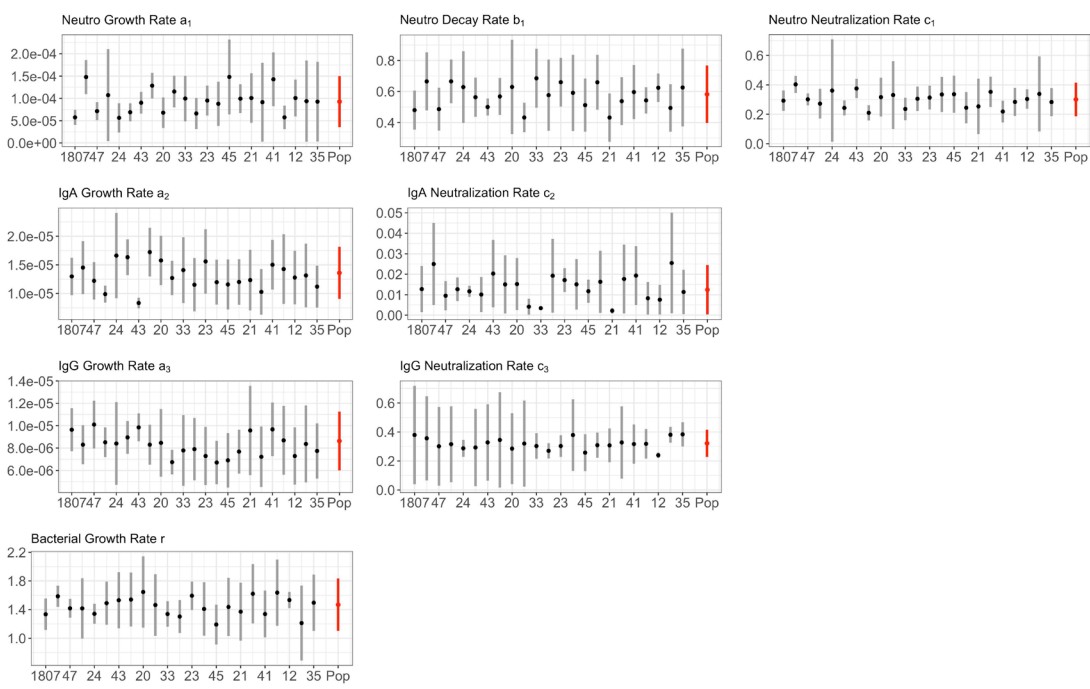

**Appendix 1—figure 3.** Parameter estimates from posterior distribution for BTG. Full details in *Appendix 1—figure 1*.

## Appendix 2

### Sensitivity analysis

Sensitivity analysis showed that the three co-infections exhibited comparable trends and were more sensitive to immune changes than *B. bronchiseptica* only rabbits (*Appendix 2—figures 1–2*); additional details in the main text.

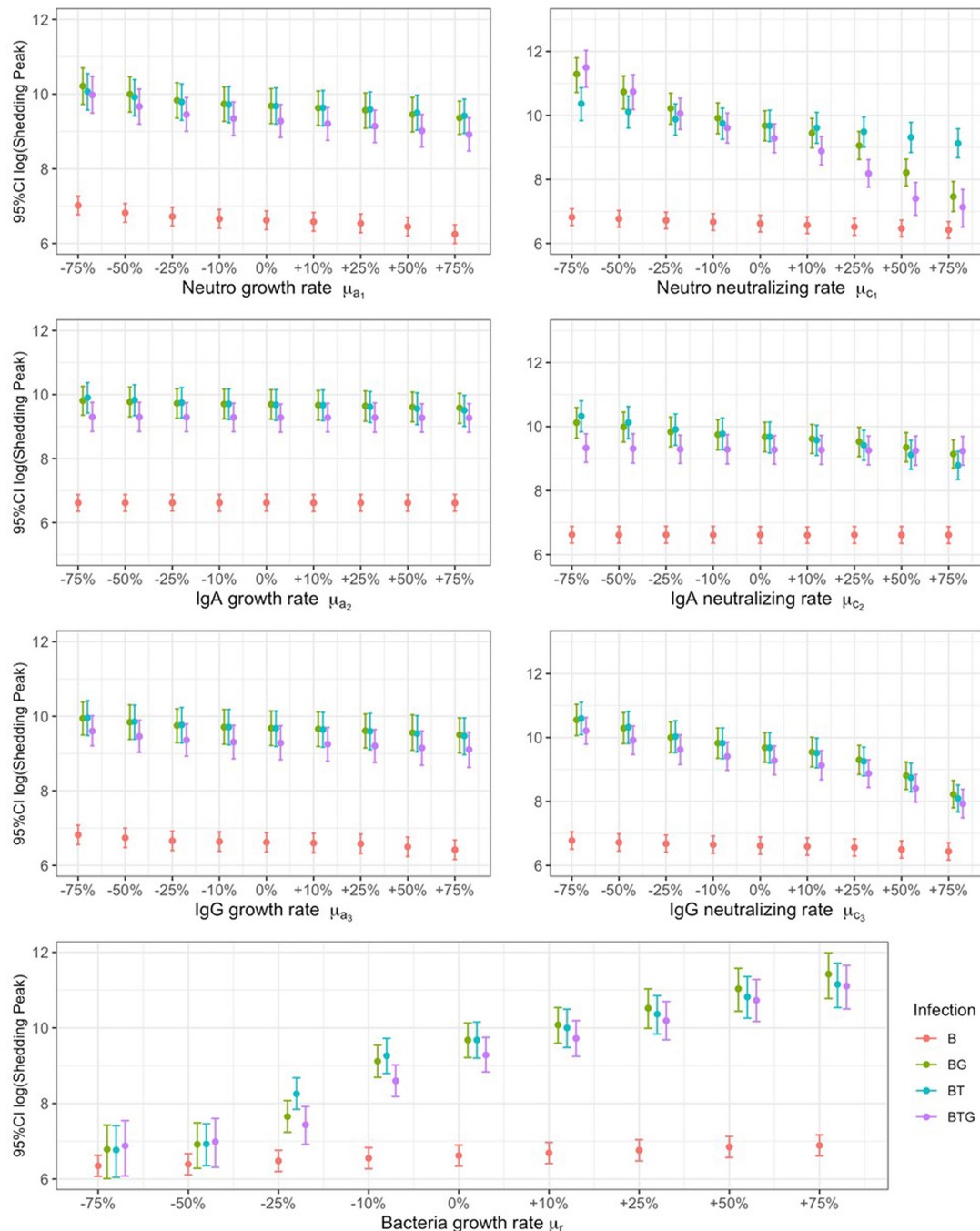

**Appendix 2—figure 1.** Relationship between *B.bronchiseptica* peak of shedding and percentile changes in neutrophils, IgA, and IgG. The growth *a*, and neutralizing *c*, rates of neutrophils (subscript 1), IgA (subscript 2) and IgG (subscript 3), including bacterial growth rate *r*, are reported by group. Mean estimates with 95% CIs are presented.

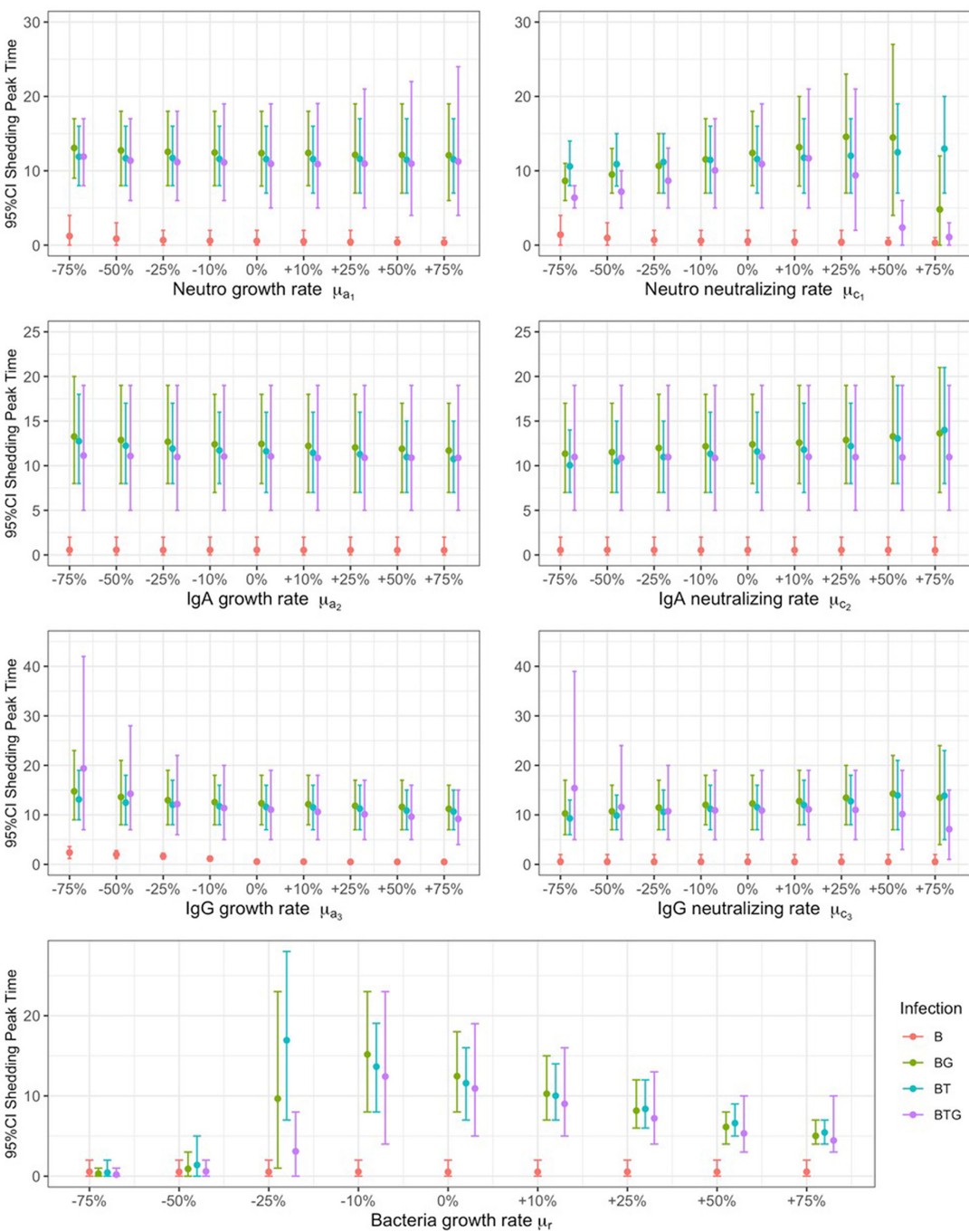

**Appendix 2—figure 2.** Relationship between *B.bronchiseptica* time to peak of shedding and percentile changes in neutrophils, IgA, and IgG. Full details in **Appendix 2—figure 1**.

# Appendix 3

## Model convergence

Here, we present convergence plots for the four model chains for $\theta_{1-10}$, where each chain was run with 100,000 iterations (*Appendix 3—figures 1–4*). For each $\theta$ and infection group the chains were well mixed and largely overlapping. For completeness, we have also included the correlations and related plots from the posterior distributions (*Appendix 3—figures 5–8*) and the scale reduction factors for each parameter (*Appendix 3—table 1*). Additional details in Materials and methods.

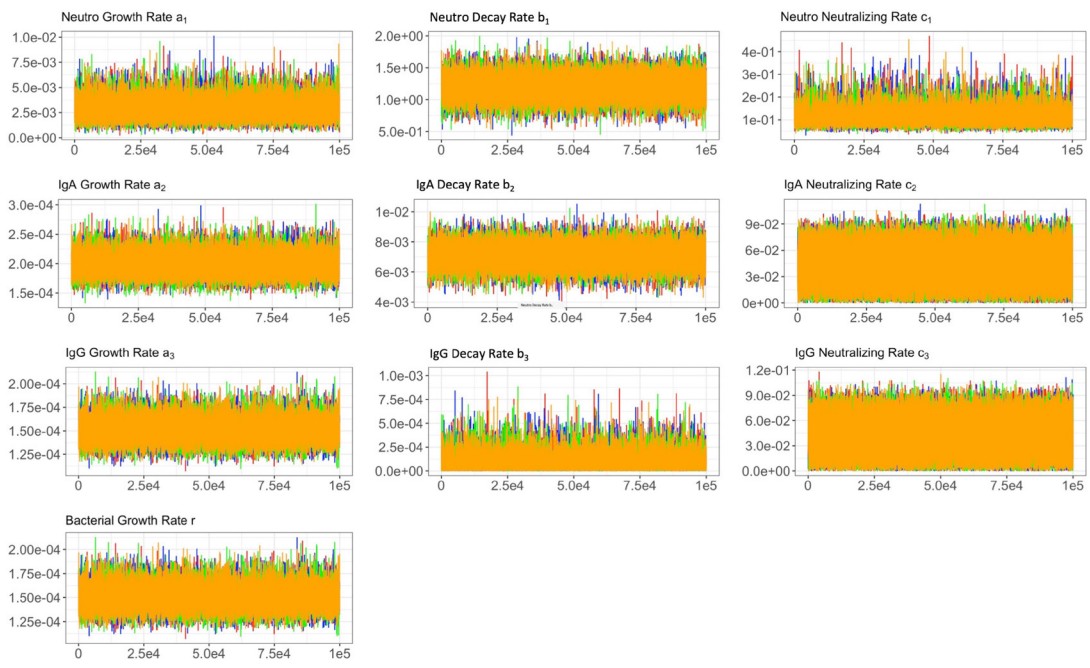

**Appendix 3—figure 1.** MCMC trace plot for B. Different colors represent the four chains.

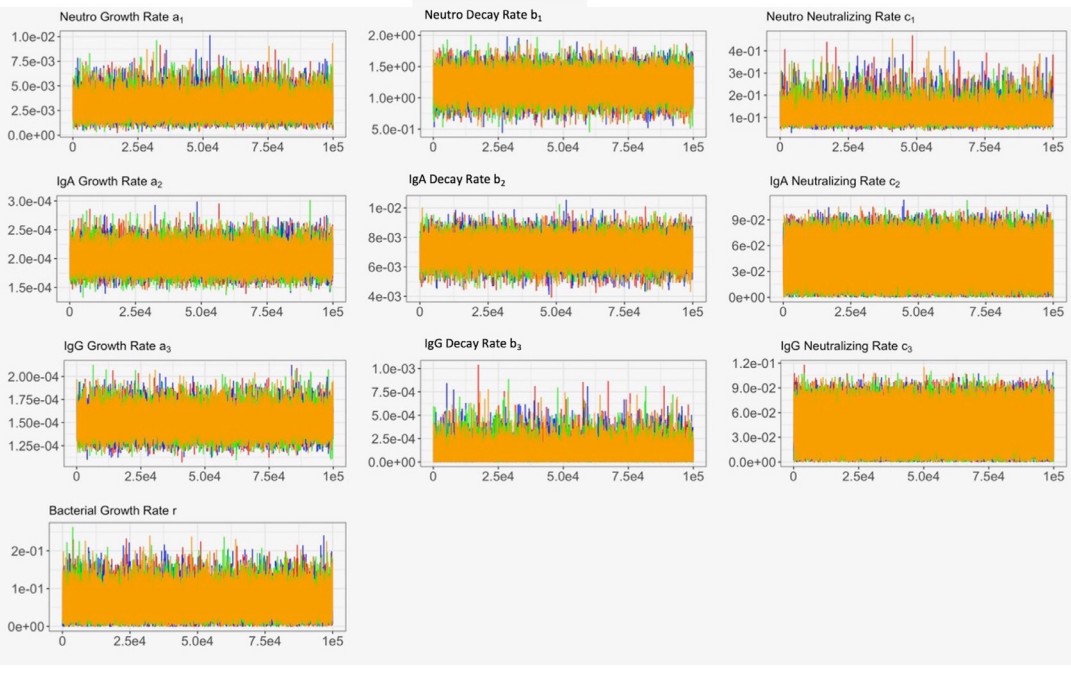

**Appendix 3—figure 2.** MCMC trace plot for BG. Different colors represent the four chains.

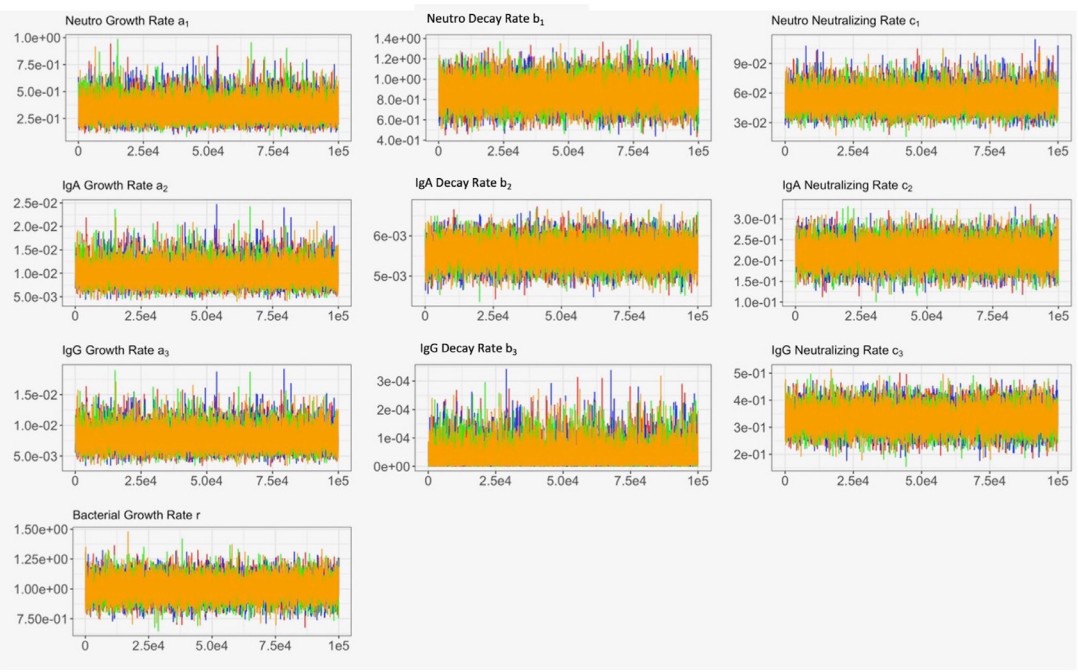

**Appendix 3—figure 3.** MCMC trace plot for BT. Different colors represent the four chains.

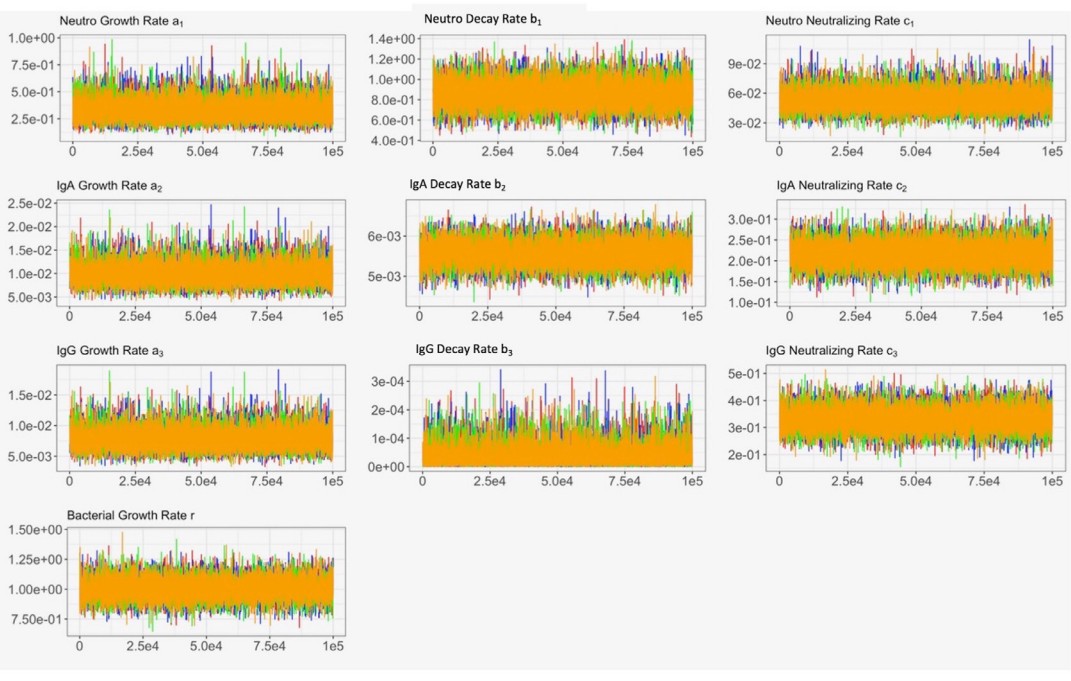

**Appendix 3—figure 4.** MCMC trace plot for BTG. Different colors represent the four chains.

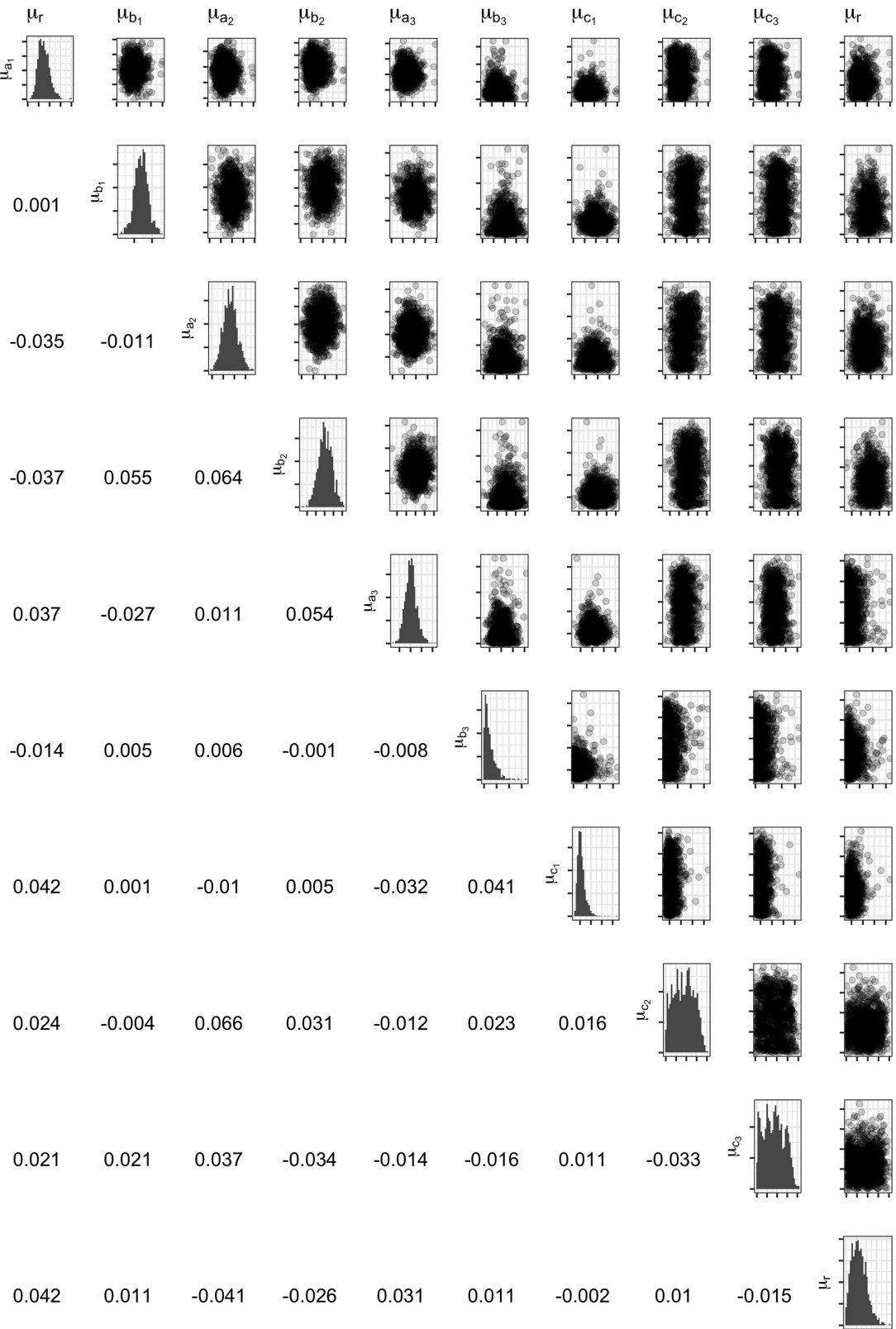

**Appendix 3—figure 5.** MCMC posteriors from B. Histograms (diagonal), pair-wise scatterplots (upper right) and related correlations (lower left) from the parameter posteriors.

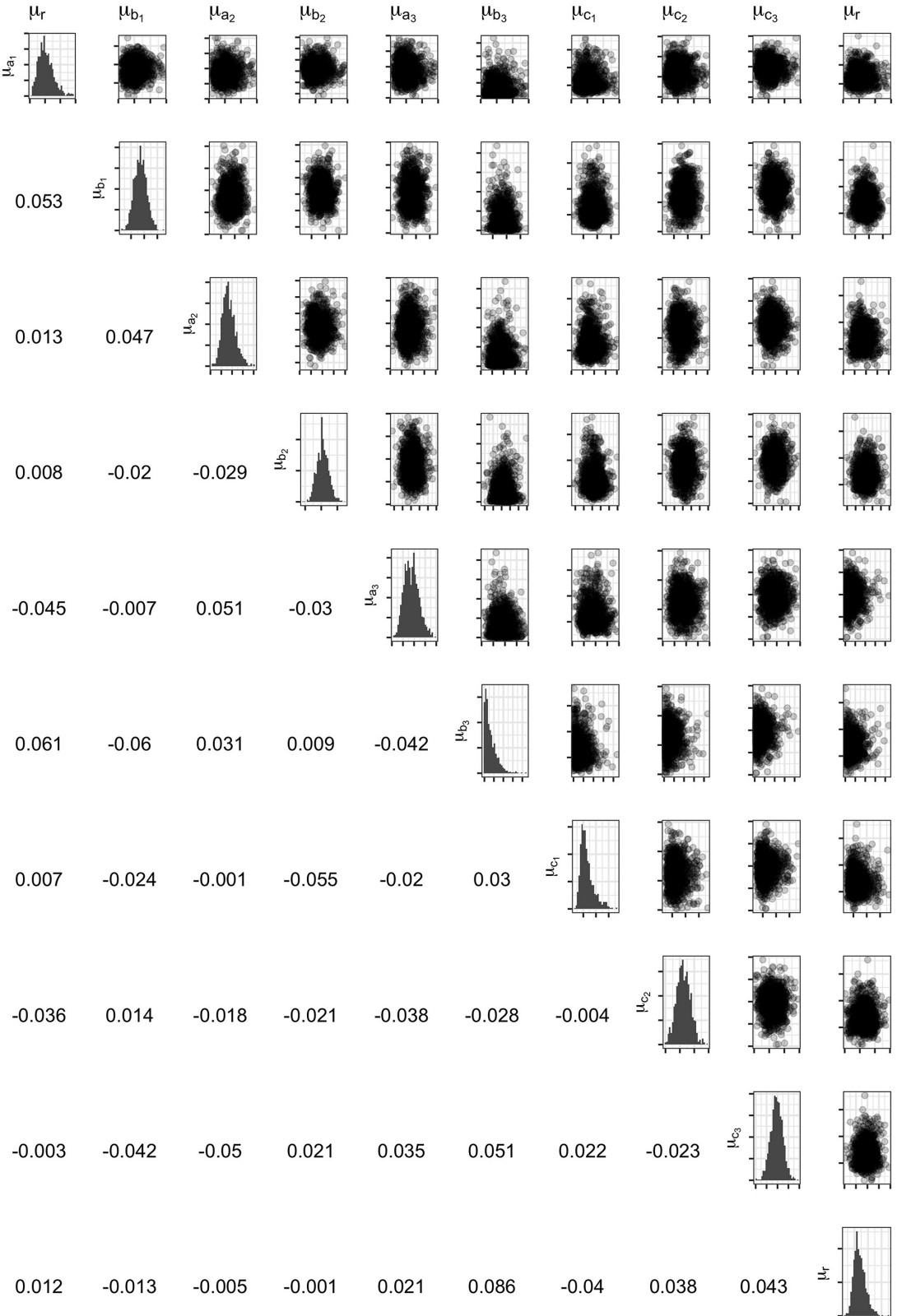

**Appendix 3—figure 6.** MCMC posteriors from BG. Histograms (diagonal), pair-wise scatterplots (upper right) and related correlations (lower left) from the parameter posteriors.

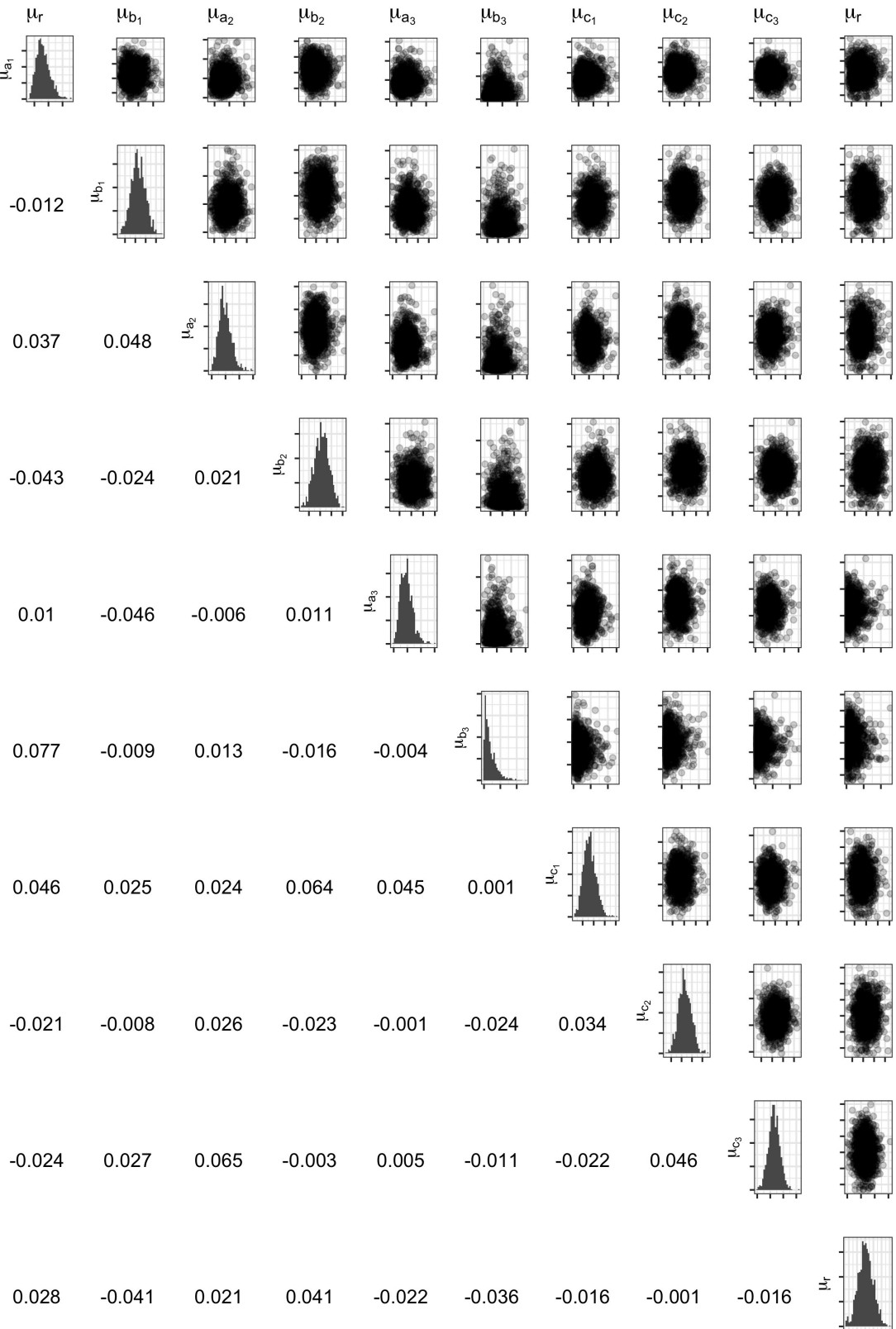

**Appendix 3—figure 7.** MCMC posteriors from BT. Histograms (diagonal), pair-wise scatterplots (upper right) and related correlations (lower left) from the parameter posteriors.

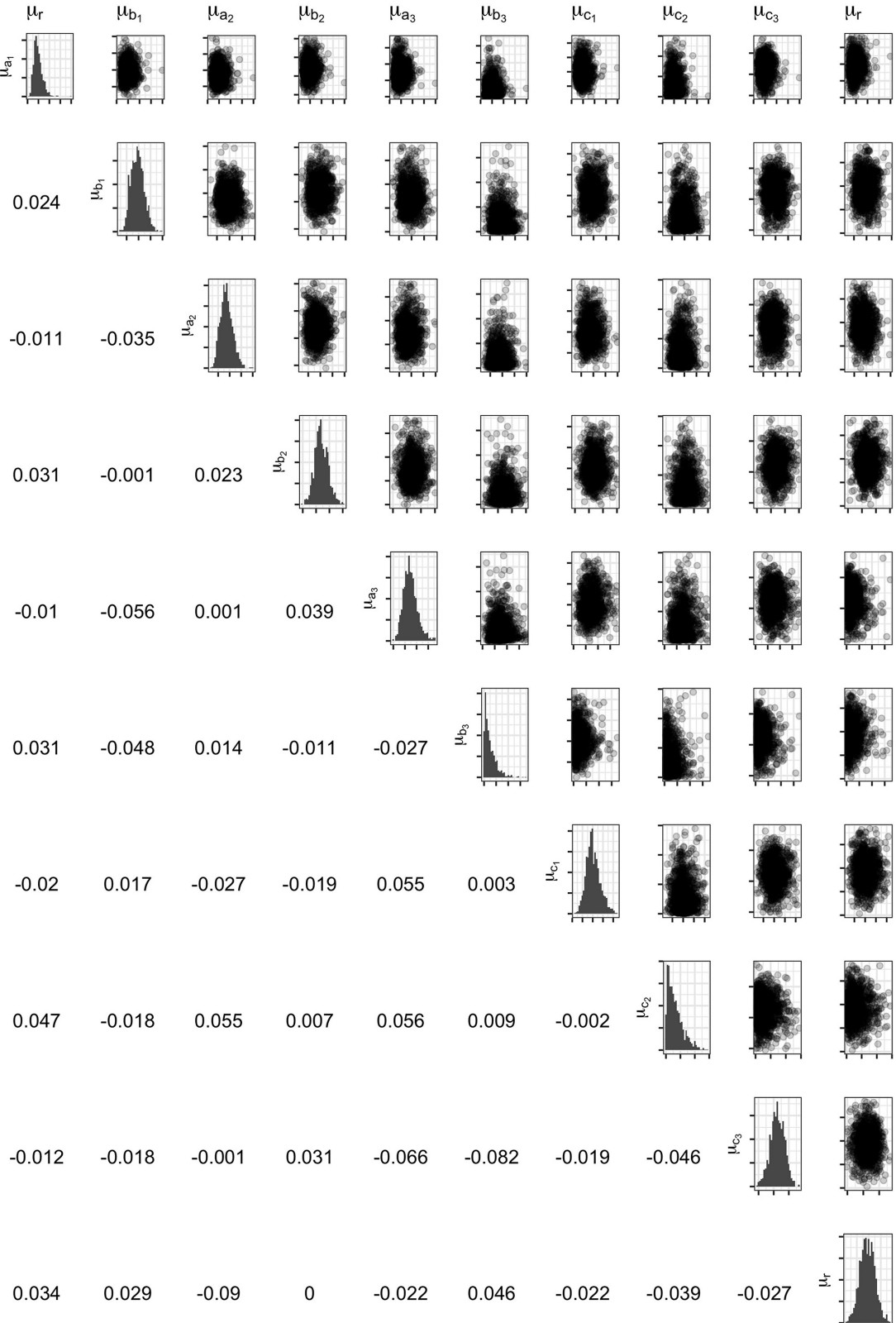

**Appendix 3—figure 8.** MCMC posteriors from BTG. Histograms (diagonal), pair-wise scatterplots (upper right) and related correlations (lower left) from the parameter posteriors.

**Appendix 3—table 1.** Scale reduction factors for each parameter of the four types of infection.

| Parameters | B | BG | BT | BTG |
|---|---|---|---|---|
| $\mu_{a1}$ | 1.000234 | 1.000103 | 1.000252 | 1.000047 |
| $\mu_{b1}$ | 1.000166 | 1.000205 | 1.000122 | 1.000073 |
| $\mu_{c1}$ | 1.000053 | 1.000047 | 0.999977 | 1.000423 |
| $\mu_{a2}$ | 1.000120 | 1.000174 | 1.000285 | 0.999986 |
| $\mu_{b2}$ | 1.000043 | 1.000049 | 1.000051 | 1.000128 |
| $\mu_{c2}$ | 1.000060 | 0.999973 | 1.000240 | 1.000316 |
| $\mu_{a3}$ | 1.000068 | 1.000166 | 1.000027 | 1.000059 |
| $\mu_{b3}$ | 1.000013 | 0.999927 | 0.999919 | 1.000089 |
| $\mu_{c3}$ | 0.999967 | 1.000016 | 1.000224 | 1.000204 |
| $\mu_{r}$ | 0.999972 | 0.999961 | 1.000139 | 1.000517 |
| log(LLH) | 1.000650 | 1.000406 | 1.000521 | 1.000401 |

## Appendix 4

### Model validation

We investigated the ability of the model to recover the correct parameter values and *Appendix 4—figure 1* shows good model performance and accuracy for every estimated parameter. Additional details in Materials and methods.

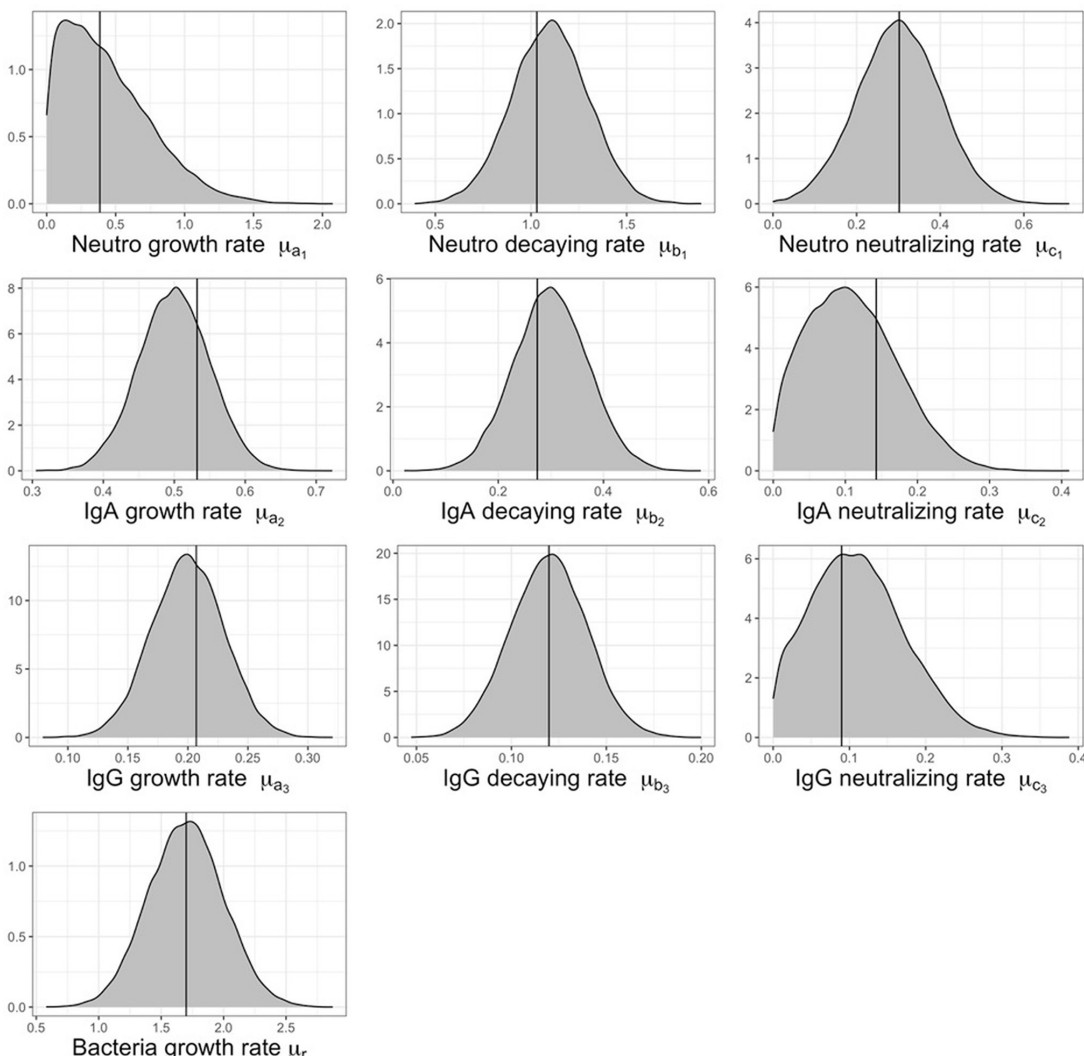

**Appendix 4—figure 1.** Model validation using group-level parameters. The vertical lines represent the true parameters that were used to simulate the datasets.

# Appendix 5

## Model selection

Three alternative dynamic models were tested: i- full model (neutrophils +IgA + IgG), ii- reduced model A (neutrophils +IgA) and iii: reduce model G (neutrophils +IgG); neutrophils were kept as the fundamental variable (*Appendix 5—table 1*). Additional details in Materials and methods.

**Appendix 5—table 1.** Bayesian Information Criterion (BIC) and level of model complexity (N), including pair-wise Δ BIC with the best fitted model.

| Models | BIC-ΔBIC | B | BG | BT | BTG | N |
|---|---|---|---|---|---|---|
| Neutrophils+IgA+IgG | BIC | -652.9 | -351.2 | 594.8 | -610.5 | 12 |
| Neutrophils+IgA | BIC | -251.3 | -302.2 | -570.9 | -507.7 | 9 |
| Neutrophils+IgA | ΔBIC | 401.5 | 48.9 | 23.9 | 102.8 | |
| Neutrophils+IgG | BIC | -314.8 | -105.6 | -442.0 | -423.9 | 9 |
| Neutrophils+IgG | ΔBIC | 338.1 | 245.6 | 152.7 | 186.5 | |

