## [Editor Report]

The authors perform experimental infections with rabbits to study how coinfection with one or more helminths affects the shedding of the respiratory bacterium *Bordetella bronchiseptica*. The results show that shedding varies strongly from one individual to the next and that co-infections with helminths lead to increased levels of shedding. The authors nicely combine within-host kinetics modelling and their longitudinal data to estimate key parameter values associated with bacterium and immune growth rates in the four conditions. These suggest that the shedding differences can be explained by differences in bacterial growth.

---

## [Decision Letter]

**Decision letter after peer review:**

Thank you for submitting your article "Gastrointestinal helminths increase *Bordetella bronchiseptica* shedding and host variation in supershedding" for consideration by *eLife*. Your article has been reviewed by 3 peer reviewers, and the evaluation has been overseen by a Reviewing Editor and Aleksandra Walczak as the Senior Editor. The following individual involved in review of your submission has agreed to reveal their identity: Samuel Alizon (Reviewer #1).

Essential revisions:

1) The authors present their results in the context of "supershedding" and, more generally, the idea that the variance in the number of secondary infections caused by an individual (i.e. R0) can matter in addition to the mean. Indeed, as popularized by Lloyd-Smith et al. (2005), even if the mean R0 is constant, an increased heterogeneity between individuals will affect disease emergence and spread. The authors appear to be presenting their work in this context and I agree that co-infections could be increasing the level of individual variation. However, in their results (and even more generally in the context of co-infections), I am not sure this is appropriate because it appears to me that the main consequence of the co-infection is to increase the mean of the distribution (rabbits shed more bacterial) rather than its heterogeneity (the proportion of rabbits that do not shed bacteria remains comparable with co-infections). If the authors really wish to keep the focus on the importance of heterogeneity, they should show that it matters and, for instance, estimate the heterogeneity parameter k of a negative binomial distribution (or a similar distribution) for the distribution of shedding rates (in Figure 3), assuming that this reflects individual R0. The other option is to focus on the mean value instead of heterogeneity, which I think can be done without loss for the integrity of the manuscript.

2) From Table 1, it appears that the authors are estimating at least 27 parameters using the model. This seems like a lot and the large confidence intervals make me wonder whether the model is identifiable. I am unsure this can be shown using likelihood profiles given the number of parameters but perhaps this could be studied by simulating datasets and calculating the mean relative error associated with the inference.

3) The authors mention model comparison in passing but given the number of parameters I think it might be worth exploring this in more details, especially since co-infections seem to be leading to similar patterns.

4) More is needed in the paragraph starting on line 309. The fact that neutrophils are produced during infections by both helminth species makes the explanation in lines 315-317 seem unconvincing. Why do the neutrophils have a different impact in the two helminth infections, and could it be related to different dynamics of helminth growth in the two species? The authors mention work in mice (Rolin et al.) that seems to show a different pattern of neutrophil dynamics and impact-should readers interpret that as merely differences between the immune responses of mice and rabbits, and if so, what differences are most likely? Possibly related to these points, it might be good to emphasize here that the model used did not account for differences in helminth infection intensity-would accounting for those differences in a future model be likely to shed light on the role of neutrophils?

5) More explanation would be helpful in a couple of places, especially regarding the dynamics in B. bronchiseptica singly infected hosts (lines 208-211). The text makes it sound like the data are just not informative as to where the peak is, but it seems clear that the peak occurred at or before the start of sampling. I was surprised that the peak was earlier for the single infections, where bacteria are thought to be replicating more slowly. Naively, I thought that with a slower replication rate, the peak would be later. How do the authors interpret that finding? Is it the case that the single infections are growing slower and also brought under control faster, resulting in an earlier peak than the double and triple infections.

6) The authors should pay particular attention to the specific comments raised as part of the public review and of course all remaining comments in a point-to-point reply.

Recommendations for the authors:

*Reviewer #1:*

lines 2-3: This sentence is a bit misleading because in Lloyd-Smith et al. the R0 is attributed to the individual.

Figure 1: The numbers on the top of each panel are unclear (I guess they refer to weeks?).

line 147-148: Is it an increase in shedding or in the likelihood to shed?

line 512-514: The writing is clear but I think the manuscript could gain in clarity by discussing a bit more within-host kinetics modelling, which is barely mentioned. Furthermore, regarding the model itself, modelling the immune response is more common now than it was 20 years ago, there are still many different models. For instance, here immune activation is assumed to depend only on parasite load. Referring to earlier models that made similar assumptions would help.

line 561-562: Do you need to make some assumptions regarding the independence of the variables to obtain the final likelihood function in equation 12?

Figure 6: It would be nice to also show in Figure 6 the within-host time series resulting from the model parameter inference and not only the "growth rates" (which are actually a bit unclear). This is already in Figure S1 but I think it would really improve the study.

Instead, showing likelihood profiles in the Appendix might be a good idea.

*Reviewer #2:*

Some modifications to the language (especially removing jargon and adding further explanation in places) would help make study accessible to a broader audience. For example, the phrase "rapid variation in individual shedding" (abstract) requires more explanation-perhaps "rapid temporal changes in individual shedding" would be clearer? Another example is line 96, where it seems strange to say that "events were null", and it might be more accessible to say that rabbits did not shed at many time points even though they were known to be infected and interacted with the petri dish (assuming I understood that correctly).

Lines 241-245 are confusing, starting with the phrase "negative bacterial growth". Assuming I'm interpreting the words in line 244 as a sharp decline in bacterial abundance, what does it mean to say that "the zero time to reach this peak were represented by the initial inoculum"?

I didn't get a lot out of Figures 7 and 8. The authors might consider moving these figures to the supplement, but regardless it would be helpful to include both parameter symbols and names/brief definitions on the x-axis labels or at the very least in the caption.

*Reviewer #3:*

Abstract, first two sentences. These sentences are a bit awkward. The authors should consider recasting them.

Abstract, "Model simulations revealed…". This makes no sense as written: simulations by themselves cannot tell us anything about the real world. Please revise this sentence to more accurately characterize the relationship between the data, the conclusions, and the model simulations.

Abstract, "…the rapid variation in individual shedding…". This sentence is very unclear.

Author Summary, line 2. Consider replacing "underline" with "underlie".

Author Summary, "experiments of rabbits together with mathematical modeling". It reads as though the experiments involved rabbits doing math!

Author Summary: "at the host level, …". This sentence is unclear. The authors should revisit it, and perhaps reconsider whether it belongs in the Author Summary.

When "type 1" and "type 2" are first introduced, they should be explained. At least, it should be made clear that "type 1" refers to Th1, etc.

ll 36ff. The connection between B. bronchiseptica and pertussis is tenuous and essentially irrelevant in this context. More generally, there is neither need nor value in this tangent.

Paragraph beginning on l 45. There are a number of facts mentioned here that are not obviously related to the authors' argument. The authors should consider whether these facts belong here. If they decide that they do belong, they should explain how.

ll 56ff. The authors describe some modeling work as if it were evidence. On the face of it, this is absurd. I recommend that they consider whether these sentences are needed, or contribute, to their study. If they conclude that they do, they should revise these sentences to put the earlier modeling work in context.

l 65. I find it strange that the authors jump from the experimental design immediately into the modeling without first describing the data that they generated.

l 72. Simulations cannot, by themselves, explain anything. Moreover, it requires great imagination to interpret the authors' model as "mechanistic".

l 80. "…every week or multiple times a week." This is most vague; the authors should be more precise.

l 91. Why was a nonparametric (Wilcoxon) test used? Does the result change if a parametric (e.g. t) test is used? Are there reasons to avoid such parametric tests? If so, what are they?

ll 93-95. Are the reported differences in median (?) shedding rate among the arms statistically significant? The broad and overlapping confidence intervals suggest not.

ll 105-111. This discussion is confusing and unclear.

ll 276-279. This is speculation, which is not in itself a problem, but it should be labeled as such.

ll 440-443. The choice of sampling times seems arbitrary. Can the authors describe the rationale behind these choices a bit more carefully.

l 449. "exemplifies" → "mimics".

ll 452-455. Explain the logistical constraints and technical difficulties.

ll 468, 470. This is an unwarranted conclusion. The authors should more carefully describe the conclusions that can be drawn from the cited study. In particular, the modeling study rests on strong assumptions about the underlying immunology.

l 568. "Weakly normal prior". "Weakness" is a relative term: the authors should describe the precise form of the priors they assume.

Figures:

Figure 1. There are several problems with this data visualization. First, it makes the "Alabama First" error, whereby the data are arranged according to an irrelevant variable (in this case, animal number within date-of-sacrifice). Second, the boxplots are not appropriate in many cases, since the data are too few or too non-normal. Third, the arbitrariness inherent in the log(1+CFU/s) metric makes it hard to interpret. I cannot confidently recommend any single visualization that will correct these problems: it will probably be necessary for the authors to experiment with, for example, simpler scatterplots, violin plots, and other approaches, before they find a more satisfactory plot or set of plots.

It may be that this figure is attempting to do too much. It seeks to convey information about the stereotypical time-course of infection and about the intra- and inter-animal variability, as well as the variation in both of these with coinfection. It might be helpful to design several figures that tackle each of the above individually.

Figure 2. This figure suffers from some of the same problems as Figure 1. In addition, there is too much cramping and overplotting to distinguish the individual-animal traces. The use of log(1+CFU/s) is problematic. Since the authors are using a zero-inflated model, it seems that the zeros don't belong here.

Finally, and at least as worryingly, the trends (smoothed curves) do not appear to represent the data at all. That is, the trends are not typical. Inasmuch as these median values are the point of contact with the models, this is quite problematic: it seems likely that even if the best-fitting models explain these averages well, they fail to represent any of the individual animals well.

Figure 3. Again, the log(1+CFU/s) metric is problematic. With a different, equally arbitrary, choice of time unit, the shape of these histograms might change appreciably.

ll 64ff. The authors appear to have made choices about the inclusion and exclusion of animals on an ad hoc basis. On its face, this raises questions about the reproducibility and reliability of their conclusions. However, it is strange that they have done so, since their zero-inflated model affords them a principled way of including all animals.

Figure S1 ought to be moved into the main text.

Figure 6. The model does not appear to do a good job in capturing the data. In particular, the individual traces and the overall trend appear to be biased downwards. This may be due to the presence of zeros in the data. If this is the case, then it is puzzling why the authors do not employ their zero-inflated model to focus attention on the non-zero data. Their choice instead to use log(1+CFU/s) is problematic in its own right as well, as discussed above.

This figure also suffers from the same problems as Figure 2: the individual data are not resolved and the individual model trajectories are too crowded.

[Editors' note: further revisions were suggested prior to acceptance, as described below.]

Thank you for resubmitting your work entitled "Gastrointestinal helminths increase *Bordetella bronchiseptica* shedding and host variation in supershedding" for further consideration by *eLife*. Your revised article has been evaluated by Aleksandra Walczak (Senior Editor) and a Reviewing Editor.

The manuscript has been improved but there are some remaining issues that need to be addressed, as outlined below:

First set of remaining issues:

Figure 1-Could the authors include an x-axis label? I assume that would be something along the lines of rabbit ID.

Figures 2, 5, and 7: could the x-axis labels be fixed so that they are all legible? Also in Figure 5, the dark blue line is difficult to see against the thinner black lines.

The paragraph in lines 244-272 could use further proofreading since some phrases are difficult to parse, including "somehow represented by the infection dose" and "shedding was not statistically significant among the three groups".

Line 322-omit "prompted" or rephrase (meaning unclear).

Second set of remaining issues:

162 onwards and 335 – the definition of supershedding raises an interesting point – as noted here, supershedding is usually thought of as the maximum shedding however duration of shedding is an important component as well. Does it change the outcome at all, if the threshold was defined in terms of the integrated shedding profile?

172 Figure 2. I realise these are complex data to represent compactly, but I find it hard to follow the individual trajectories (e.g. to determine if single infections are consistently different from each other). Perhaps a few example trajectories in different colours might help?

Line 191 – Figure 3. It is hard to tell, but it looks like the negative binomial is a reasonable representation of the BT infections but that there may be greater systematic biases for the BG and BTG cases. Is this true and if so, statistically significant?

448 – I would be slightly more reserved about making a definitive statement on the impact of measured super-shedding in an experimental setting, an undefined use of 'contact' and the probability of it resulting in infection and/or a greater number of infections. There is a good chance they are right, but that's not quite the same as proving it.

684 and 932 onwards. Very good to see the convergence plots – it would be helpful also have the scale reduction factors, as its difficult see in the plots what the individual chains are actually doing – based on the text it should be fine, but having the results recorded would be useful.

902 – Good to see figures showing posterior estimates here – I much prefer them to tables. I do generally find it helpful to have figures showing the posterior distribution, preferably in correlation plots to show at least some of the interdependencies between parameters which is particularly important given the number of parameters involved. I realise it would add considerably to the appendix to show more detail for each individual, however, so long as it doesn't misrepresent the variation between individuals, I think an averaged correlation plot across all posteriors would be helpful.

---

## [Author Response]

Essential revisions:1) The authors present their results in the context of "supershedding" and, more generally, the idea that the variance in the number of secondary infections caused by an individual (i.e. R0) can matter in addition to the mean. Indeed, as popularized by Lloyd-Smith et al. (2005), even if the mean R0 is constant, an increased heterogeneity between individuals will affect disease emergence and spread. The authors appear to be presenting their work in this context and I agree that co-infections could be increasing the level of individual variation. However, in their results (and even more generally in the context of co-infections), I am not sure this is appropriate because it appears to me that the main consequence of the co-infection is to increase the mean of the distribution (rabbits shed more bacterial) rather than its heterogeneity (the proportion of rabbits that do not shed bacteria remains comparable with co-infections). If the authors really wish to keep the focus on the importance of heterogeneity, they should show that it matters and, for instance, estimate the heterogeneity parameter k of a negative binomial distribution (or a similar distribution) for the distribution of shedding rates (in Figure 3), assuming that this reflects individual R0. The other option is to focus on the mean value instead of heterogeneity, which I think can be done without loss for the integrity of the manuscript.

This is a very good point and important in the broader discussion on co-infection. We want to clarify that we do not quantify the proportion of hosts that shed or not shed, they all shed at some point during the experiment (the seven rabbits that never shed were excluded from the analysis).

This study investigated variation in the amount and frequency of shedding events at three levels: (i) between four types of infection, (ii) between hosts within each type of infection and iii- by every individual over time.

We have now clarified the objective of this work and in our updated results we show that helminths increase the level of bacteria shed, the frequency of shedding events (i.e. events when rabbits do not shed is 56% in the B group and 40%-25% in the co-infected groups) and variation in the magnitude of these events (Negative binomial k, BG: 0.37, BT: 0.45 and BTG: 0.21, no fitting for the B group because only three data-bins are available). This indicates that helminths affect both the mean of bacteria shed and variation in this mean, including the emergence of supershedding events.

We have revised figure 3 (the original plot was misleading) and we have now: i- used the same interval size (CFU/s=0.5) to classify shedding for the four infections, ii- provided the parameter k from fitting a Negative Binomial, iii- updated the 95th and 99th percentile thresholds as estimated using the whole data set (four infections together), and iv- presented the data as frequency of CFU/s events (i.e. not log-transformed as originally showed).

2) From Table 1, it appears that the authors are estimating at least 27 parameters using the model. This seems like a lot and the large confidence intervals make me wonder whether the model is identifiable. I am unsure this can be shown using likelihood profiles given the number of parameters but perhaps this could be studied by simulating datasets and calculating the mean relative error associated with the inference.

We are not sure how the Reviewer has calculated this number. The number of shared parameters within each infection group is 18 as given by equation (9) in the main text, while the number of individual parameters, given by equation (10), varies depending on the number of rabbits for each infection. To provide robust support to our approach, we have followed the advice and have now created new sections in Material and Methods (‘Model validation’) and Appendix (Appendix-4) where we investigate the ability of the model to recover the correct parameter values and show good model performance and accuracy for every estimated parameter.

3) The authors mention model comparison in passing but given the number of parameters I think it might be worth exploring this in more details, especially since co-infections seem to be leading to similar patterns.

We have now created a new section in Material and Methods (‘Model Selection’) and in the

Appendix (Appendix-5) where we compare three model formulations: the full model

(neutrophils+IgA+IgG) and two simplified versions (neutrophils+IgA and neutrophils+IgG). The full model was selected as the best compromise between goodness of fit and parsimony, and subsequently used in our analysis.

4) More is needed in the paragraph starting on line 309. The fact that neutrophils are produced during infections by both helminth species makes the explanation in lines 315-317 seem unconvincing. Why do the neutrophils have a different impact in the two helminth infections, and could it be related to different dynamics of helminth growth in the two species? The authors mention work in mice (Rolin et al.) that seems to show a different pattern of neutrophil dynamics and impact-should readers interpret that as merely differences between the immune responses of mice and rabbits, and if so, what differences are most likely? Possibly related to these points, it might be good to emphasize here that the model used did not account for differences in helminth infection intensity-would accounting for those differences in a future model be likely to shed light on the role of neutrophils?

We have carefully revised this section and removed some of the inconsistencies that caused confusion. We now discuss more coherently our neutrophil results and how our findings relate to previous studies in mice using Bordetella and other pathogens. To stress that we did not explicitly include the dynamics of the two helminths, we have included comments in the Introduction, Material and Methods and Results.

5) More explanation would be helpful in a couple of places, especially regarding the dynamics in B. bronchiseptica singly infected hosts (lines 208-211). The text makes it sound like the data are just not informative as to where the peak is, but it seems clear that the peak occurred at or before the start of sampling. I was surprised that the peak was earlier for the single infections, where bacteria are thought to be replicating more slowly. Naively, I thought that with a slower replication rate, the peak would be later. How do the authors interpret that finding? Is it the case that the single infections are growing slower and also brought under control faster, resulting in an earlier peak than the double and triple infections.

We have revised this section and provided more clarity on the dynamics of shedding in B. bronchiseptica only rabbits. Briefly, we note that model simulations place the peak right at the start of the trial, however, this should be taken with caution, since there are no shedding data to train the model during these first ten days (as noted in Material and Methods we used a different substrate for bacteria shedding that did not work). We also note that while we still report the model prediction at this early time this needs to be eventually validated in the laboratory.

Recommendations for the authors:Reviewer #1:lines 2-3: This sentence is a bit misleading because in Lloyd-Smith et al. the R0 is attributed to the individual.

The reviewer is correct in that Ro for microparasites is the number of secondary infections generated by a primary case; however, while this is the unit of measurement, essentially, Ro is a measurement of pathogen/parasite fitness and its ability to be transmitted.

Figure 1: The numbers on the top of each panel are unclear (I guess they refer to weeks?).

As originally noted in the legend: “The total number of rabbits monitored in each infection is reported in parenthesis at the top left of each plot”. To avoid confusion this detail has now been removed and the legend has been revised.

line 147-148: Is it an increase in shedding or in the likelihood to shed?

This is an increase in the probability of becoming supershedders (whether this is at the 95% or 99% percentile threshold), the sentence has been clarified.

line 512-514: The writing is clear but I think the manuscript could gain in clarity by discussing a bit more within-host kinetics modelling, which is barely mentioned. Furthermore, regarding the model itself, modelling the immune response is more common now than it was 20 years ago, there are still many different models. For instance, here immune activation is assumed to depend only on parasite load. Referring to earlier models that made similar assumptions would help.

We now briefly discuss previous models that investigated the within-host interaction between Bordetella and the immune response, and how our model differs from those; we also provide additional references of models that used assumptions similar to our work.

line 561-562: Do you need to make some assumptions regarding the independence of the variables to obtain the final likelihood function in equation 12?

We have clarified that there is the assumption of independence between the variables in the likelihood calculations. However, we also note that the variables have an interactive relationship that are captured in the dynamical models. Therefore, the assumption of independence inside the likelihood function is a plausible one.

Figure 6: It would be nice to also show in Figure 6 the within-host time series resulting from the model parameter inference and not only the "growth rates" (which are actually a bit unclear). This is already in Figure S1 but I think it would really improve the study.Instead, showing likelihood profiles in the Appendix might be a good idea.

We think the reviewer refers to the original figure 5, the (original) figure 6 described bacterial shedding from the experimental and simulation data and included both individual and median results. We have updated figure 5 (current figure 6) and included the individual time series for the neutralization rates of the co-infection groups. We have not included individual time series for the growth and decay rates because of the very narrow CIs, which limit the ability to disentangle each individual trend. Simulations for the B group were done only at the group level. As advised by reviewer #3, we have now moved the original figure S1 in the main text (currently figure 5) to show the accuracy of model inference to experimental data.

Reviewer #2:Some modifications to the language (especially removing jargon and adding further explanation in places) would help make study accessible to a broader audience. For example, the phrase "rapid variation in individual shedding" (abstract) requires more explanation-perhaps "rapid temporal changes in individual shedding" would be clearer? Another example is line 96, where it seems strange to say that "events were null", and it might be more accessible to say that rabbits did not shed at many time points even though they were known to be infected and interacted with the petri dish (assuming I understood that correctly).

We have revised the whole manuscript to improve clarify and accessibility to a broader audience.

Lines 241-245 are confusing, starting with the phrase "negative bacterial growth". Assuming I'm interpreting the words in line 244 as a sharp decline in bacterial abundance, what does it mean to say that "the zero time to reach this peak were represented by the initial inoculum"?

We agree that this oxymoron is confusing and have revised the paragraph.

I didn't get a lot out of Figures 7 and 8. The authors might consider moving these figures to the supplement, but regardless it would be helpful to include both parameter symbols and names/brief definitions on the x-axis labels or at the very least in the caption.

We have followed the Reviewer’s suggestion and moved these figures in the Appendix (now Appendix 2- figures 1-2); we also revised the labeling as advised to increase clarity.

Reviewer #3:Abstract, first two sentences. These sentences are a bit awkward. The authors should consider recasting them.Abstract, "Model simulations revealed…". This makes no sense as written: simulations by themselves cannot tell us anything about the real world. Please revise this sentence to more accurately characterize the relationship between the data, the conclusions, and the model simulations.Abstract, "…the rapid variation in individual shedding…". This sentence is very unclear.

The abstract has been revised to improve clarity.

Author Summary, line 2. Consider replacing "underline" with "underlie".Author Summary, "experiments of rabbits together with mathematical modeling". It reads as though the experiments involved rabbits doing math!Author Summary: "at the host level, …". This sentence is unclear. The authors should revisit it, and perhaps reconsider whether it belongs in the Author Summary.

Accordingly and as noted above, the summary has been revised.

When "type 1" and "type 2" are first introduced, they should be explained. At least, it should be made clear that "type 1" refers to Th1, etc.

We have replaced type 1 and type 2 with Th1 and Th2, which are terms more commonly known. When first mentioned, we refer to ‘Th1 inflammatory response’ or ‘Th2 anti-inflammatory response’ to provide more information on the type of response.

ll 36ff. The connection between B. bronchiseptica and pertussis is tenuous and essentially irrelevant in this context. More generally, there is neither need nor value in this tangent.

We included this connection as relevant in the context of understanding the dynamics of shedding of Bordetella species, including in humans where whooping cough remains an infection of serious concern. In this context, the connection with humans gives broad relevance to the study beyond the rabbit system.

Paragraph beginning on l 45. There are a number of facts mentioned here that are not obviously related to the authors' argument. The authors should consider whether these facts belong here. If they decide that they do belong, they should explain how.

This section discusses co-infection and provides a background on what has been previously done on co-infections with B. bronchiseptica and helminths. This is also important to understand the system and the rationale of our study, and to prepare the reader to the following section on the objective and general working approach. We have revised the Introduction and this section accordingly.

ll 56ff. The authors describe some modeling work as if it were evidence. On the face of it, this is absurd. I recommend that they consider whether these sentences are needed, or contribute, to their study. If they conclude that they do, they should revise these sentences to put the earlier modeling work in context.

In this section we describe findings from our previous work using experimental data and modeling on the same system that we use in the current study. This is relevant for two reasons: first, it provides useful information to understand the system and second, it informs on a possible mechanism of immune regulation of Bordetella in rabbits. This section has been revised and references on similar work in mice added. Please, see additional previous comment regarding the need of this whole paragraph.

l 65. I find it strange that the authors jump from the experimental design immediately into the modeling without first describing the data that they generated.

We have now revised this section to avoid the ‘jump from experiments into modeling’ and provided more information on the data used, including referring to previous work.

l 72. Simulations cannot, by themselves, explain anything. Moreover, it requires great imagination to interpret the authors' model as "mechanistic".

We have revised the sentence accordingly, however, we disagree with the reviewer regarding the term ‘mechanistic’ as used in this context. The model we propose offers a parsimonious, and simplified, explanation of the mechanism in which the three immune variables affect the temporal dynamics of Bordetella infection and shedding (please, see further comments below).

l 80. "…every week or multiple times a week." This is most vague; the authors should be more precise.

We have now clarified that shedding data were collected every week or two-three times a week for every rabbit and refer to Material and Methods (Bacteria shed enumeration) for additional details.

l 91. Why was a nonparametric (Wilcoxon) test used? Does the result change if a parametric (e.g. t) test is used? Are there reasons to avoid such parametric tests? If so, what are they?

Thank you for pointing this out. We found a make a mistake in the initial choice of the hypothesis test. The appropriate test to use is the Kruskal Wallis test, which we used to compare the group medians among the four types of infection with Bonferroni’s correction for multiple testing. We choose a non-parametric test because the data are highly aggregated (see Figure 3).

ll 93-95. Are the reported differences in median (?) shedding rate among the arms statistically significant? The broad and overlapping confidence intervals suggest not.

The reviewer is correct, we found no differences between the co-infected groups and included the significant term (p>0.05) of our analysis to stress our point.

ll 105-111. This discussion is confusing and unclear.

This section has been revised to avoid confusion.

ll 276-279. This is speculation, which is not in itself a problem, but it should be labeled as such.

We have revised the section to avoid definitive conclusions.

ll 440-443. The choice of sampling times seems arbitrary. Can the authors describe the rationale behind these choices a bit more carefully.

The experimental design is driven by the dynamics/life cycles of the three infectious agents. Specifically, we followed the standard design of Bordetella infection in mice, which considers important phases in the bacteria-host interaction (e.g. Kirimanjeswara et al. 2003). We also included additional time points to increase accuracy in our dataset and representative phases in the life cycle of the two helminths (Murphy et al. 2022, 2013, Cattadori et al. 2019) that might be important in affecting Bordetella. We have revised the sentence and included appropriate references to support our sampling design.

l 449. "exemplifies" → "mimics".

We made the change as suggested.

ll 452-455. Explain the logistical constraints and technical difficulties.

We consider this information irrelevant to the Methods, however, we have positively addressed this comment as follow: “The use of a different number of hosts and frequency of sampling was determined by logistical constraints (i.e. personnel availability).”

ll 468, 470. This is an unwarranted conclusion. The authors should more carefully describe the conclusions that can be drawn from the cited study. In particular, the modeling study rests on strong assumptions about the underlying immunology.

While we do agree that ‘the modeling study rests on strong assumptions about the underlying immunology’, the cited models do indeed build on data from laboratory experiments and current knowledge on the host immune response to Bordetella infection. Our modeling assumptions follow a similar rationale using data from our laboratory experiments. We have revised the sentence to avoid definitive statements.

l 568. "Weakly normal prior". "Weakness" is a relative term: the authors should describe the precise form of the priors they assume.

We have now provided the prior values in table 2 and noted that they were generated from a Normal distribution.

Figures:Figure 1. There are several problems with this data visualization. First, it makes the "Alabama First" error, whereby the data are arranged according to an irrelevant variable (in this case, animal number within date-of-sacrifice). Second, the boxplots are not appropriate in many cases, since the data are too few or too non-normal. Third, the arbitrariness inherent in the log(1+CFU/s) metric makes it hard to interpret. I cannot confidently recommend any single visualization that will correct these problems: it will probably be necessary for the authors to experiment with, for example, simpler scatterplots, violin plots, and other approaches, before they find a more satisfactory plot or set of plots.It may be that this figure is attempting to do too much. It seeks to convey information about the stereotypical time-course of infection and about the intra- and inter-animal variability, as well as the variation in both of these with coinfection. It might be helpful to design several figures that tackle each of the above individually.

Figure 1 has been designed to mimic our experiential design, a stereotypical visualization yet an effective way to convey a simple message while avoiding graphic redundancy. Within each infection rabbits are ordered from left to right following the experimental time. We sacrificed groups of 4 rabbits at a time, groups sampled early on the left and groups sampled late on the right; to facilitate the vision of the time course we have reported the week of sampling for each group of animals. The order of these four rabbits within each group is arbitrary since this is the only time we report the ID of the host. We tried other approaches and the boxplot was still the best compromise between a representation of the experimental design, which we wanted to provide, and general information on the type of infections and metrics of each host. As previously reported, the log(CFU/s+1) is a useful choice for the visualization of highly variable data.

Figure 2. This figure suffers from some of the same problems as Figure 1. In addition, there is too much cramping and overplotting to distinguish the individual-animal traces. The use of log(1+CFU/s) is problematic. Since the authors are using a zero-inflated model, it seems that the zeros don't belong here.Finally, and at least as worryingly, the trends (smoothed curves) do not appear to represent the data at all. That is, the trends are not typical. Inasmuch as these median values are the point of contact with the models, this is quite problematic: it seems likely that even if the best-fitting models explain these averages well, they fail to represent any of the individual animals well.

Figure 2 represents the experimental longitudinal data on shedding and highlights the high heterogeneity in the shedding events both within and between hosts. The smoothed individual curves are challenged by the high variation between consecutive time points, this is also highlighted in the median trend. To help with the visualization, data were transformed to log(CFU/s+1), as mentioned above. We have revised this figure and now plot the individual trajectories by simply joining the events with straight segments but keeping the smoothed median fitted to the group data. We have also revised the legend and our comments to this figure in the main text. We tried alternative plotting but visualization did not improve, given the nature of the data and what we wanted to convey.

Figure 3. Again, the log(1+CFU/s) metric is problematic. With a different, equally arbitrary, choice of time unit, the shape of these histograms might change appreciably.

The frequency distribution is now based on CFU/s data (not log-transformed) with intensity of shedding grouped in classes of 0.5 unit intervals (these are not time intervals), plus the initial class 0 for the ‘null-shedding’. This approach has been applied to each type of infection. We have also calculated the 99th and 95th percentile thresholds from the whole data set and used these cut-offs to estimate the number of supershedding events/hosts in each infection group. Finally, we have fitted a Negative Binomial to the frequency distribution. This updated figure provides new results that are reported and discussed in the main text.

ll 64ff. The authors appear to have made choices about the inclusion and exclusion of animals on an ad hoc basis. On its face, this raises questions about the reproducibility and reliability of their conclusions. However, it is strange that they have done so, since their zero-inflated model affords them a principled way of including all animals.

The reviewer probably meant l87 of the original manuscript. To avoid confusion, we have now clarified that since we were interested in the dynamics of shedding, rabbits that never shed (n = 7) were excluded from the subsequent analysis. This number of rabbits is also too small to perform any meaningful analysis on the immune response associated with rabbits that do not shed. Please see detailed comments on the use of animals above.

Figure S1 ought to be moved into the main text.

The figure has now been moved into the main text as suggested.

Figure 6. The model does not appear to do a good job in capturing the data. In particular, the individual traces and the overall trend appear to be biased downwards. This may be due to the presence of zeros in the data. If this is the case, then it is puzzling why the authors do not employ their zero-inflated model to focus attention on the non-zero data. Their choice instead to use log(1+CFU/s) is problematic in its own right as well, as discussed above.This figure also suffers from the same problems as Figure 2: the individual data are not resolved and the individual model trajectories are too crowded.

We have revised the quality of figure 6 (now figure 7) to make our results clearer. We also realized that our comments were not completely accurate and, thus, the main text has now been revised. Briefly, we note that “simulations of individual time series captured relatively well the general trend and the high level of shedding early in the infection, particularly for BTG and BT, and less clearly for BG”. By changing the individual trajectories to a black tone, we can now see the general good fit to the data.

We did apply the zero-inflated relationship between shedding and infection and used the logtransformed data to reduce variation and skewed trend. Specifically, shedding data were modeled with a zero-inflated distribution where the 0 values constitute the zero-inflated part and the non-zero values are captured by a log-Normal distribution. Please, see our previous comments on the use of log-transformed data.

[Editors' note: further revisions were suggested prior to acceptance, as described below.]

The manuscript has been improved but there are some remaining issues that need to be addressed, as outlined below:First set of remaining issues:Figure 1-Could the authors include an x-axis label? I assume that would be something along the lines of rabbit ID.

Good point, we have now included the x-label “rabbit ID’ as suggested.

Figures 2, 5, and 7: could the x-axis labels be fixed so that they are all legible? Also in Figure 5, the dark blue line is difficult to see against the thinner black lines.

We have revised the figures and provided a clearer format by adjusting the x-axis and making the contrast between lines clearer.

The paragraph in lines 244-272 could use further proofreading since some phrases are difficult to parse, including "somehow represented by the infection dose" and "shedding was not statistically significant among the three groups".

We have revised the section to avoid confusion.

Line 322-omit "prompted" or rephrase (meaning unclear).

The sentence has been revised accordingly.

Second set of remaining issues:162 onwards and 335 – the definition of supershedding raises an interesting point – as noted here, supershedding is usually thought of as the maximum shedding however duration of shedding is an important component as well. Does it change the outcome at all, if the threshold was defined in terms of the integrated shedding profile?

This is an interesting consideration. We have further investigated the shedding data by calculating the area under the curve for each individual trajectory as composite trapezoids of the area under the linear lines connecting all the shedding values and the x-axis. We found that at a 95% threshold (defined from all infections) the supershedding events were as follow, B: 0, BG: 1 (4.5%), BT: 0 and BTG: 3 (12.5%), while at a 99% only the BG group (1, 4.5%) continued to have supershedding events. The high BG supershedders seem to substantially drive this pattern and affect the other groups. While interesting, we did not include this result in the manuscript to avoid confusion with the more classical approach we present and the general message we wanted to convey.

172 Figure 2. I realise these are complex data to represent compactly, but I find it hard to follow the individual trajectories (e.g. to determine if single infections are consistently different from each other). Perhaps a few example trajectories in different colours might help?

We have revised the figure and made it in a bigger size, we have also highlighted the trajectories of a few individuals for every infection, as suggested.

Line 191 – Figure 3. It is hard to tell, but it looks like the negative binomial is a reasonable representation of the BT infections but that there may be greater systematic biases for the BG and BTG cases. Is this true and if so, statistically significant?

We have now noted that the negative binomial distribution was not significantly different from the frequency of each co-infection dataset (p>0.05). In the original version we also provided the CIs of the aggregation parameter k.

448 – I would be slightly more reserved about making a definitive statement on the impact of measured super-shedding in an experimental setting, an undefined use of 'contact' and the probability of it resulting in infection and/or a greater number of infections. There is a good chance they are right, but that's not quite the same as proving it.

We have revised the section to avoid an overstatement in our reasoning.

684 and 932 onwards. Very good to see the convergence plots – it would be helpful also have the scale reduction factors, as its difficult see in the plots what the individual chains are actually doing – based on the text it should be fine, but having the results recorded would be useful.

We have now included Table 1 in Appendix 3 where we report the scale reduction factors of each parameter for the four types of infections.

902 – Good to see figures showing posterior estimates here – I much prefer them to tables. I do generally find it helpful to have figures showing the posterior distribution, preferably in correlation plots to show at least some of the interdependencies between parameters which is particularly important given the number of parameters involved. I realise it would add considerably to the appendix to show more detail for each individual, however, so long as it doesn't misrepresent the variation between individuals, I think an averaged correlation plot across all posteriors would be helpful.

We have included in Appendix 3 the correlations and related plots of the posteriors for every infection group, as advised.